# The Use of the LITMUS Quasi-Universal Nonword Repetition Task to Identify DLD in Monolingual and Early Second Language Learners Aged 8 to 10

Angela Grimm

Institute of Linguistics, Goethe University Frankfurt, 60323 Frankfurt, Germany; a.grimm@em.uni-frankfurt.de

**Abstract:** This study evaluates whether the short version of the German LITMUS quasi-universal nonword repetition task (LITMUS-QU-NWR) can be used as an index test for monolingual and early second language learners (eL2) of German aged 8 to 10 years. The NWR taps into quasi-universal phonological knowledge via the so-called language-independent part and into language-specific phonological knowledge via the language-dependent part. Thirty-six monolingual and thirty-three eL2 learners of German, typically developing (TD) and diagnosed as language-impaired (DLD), participated in the study. The effects of the language group (Mo vs. eL2) and the clinical status (TD vs. DLD) on repetition accuracy are investigated by a logistic mixed-model analysis. Receiver operating characteristics (ROC) and likelihood ratios are calculated to determine the diagnostic accuracy of the two parts. The group comparisons showed significant effects of the clinical status but not of the language group. The ROC analyses and the likelihood ratios reveal better diagnostic values for the language-dependent compared to the language-independent part and almost similar diagnostic values for the monolingual and the eL2 group. The results indicate that the LITMUS-QU-NWR helps to disentangle DLD and DLD in monolingual children and eL2 learners aged 8 to 10 years.

**Keywords:** nonword repetition; monolingual children; early second language learners; German; DLD; diagnostic accuracy; LITMUS-QU-NWR (German); phonological complexity

---

## 1. Introduction

Numerous studies have shown that nonword repetition (NWR) provides a reliable indicator of language impairments in monolingual children (Graf Estes et al. 2007 for a meta-analysis of English NWR tests; Schuchardt et al. 2013 for German; Schwob et al. 2021 for a meta-analysis of NWR tests in different languages). For monolingual children, the meta-analysis of Graf Estes et al. (2007) reveals that word-like, longer, and articulatory complex nonwords better discriminate between TD and DLD compared to less word-like, short, and articulatory simpler nonwords. Moreover, a low phonotactic probability increases the level of difficulty for children with language impairments. Furthermore, Graf Estes et al. (2007) find consistently good differentiation of nonword repetition independently of the children's age. Trisyllabic and longer items discriminate better than shorter items. Mono- and bisyllabic items discriminate if big samples are examined or if long item lists are used ('short item effect'). According to Graf Estes et al. (2007), the short item effect shows that nonword repetition does not only tap into phonological working memory (contra Archibald and Gathercole 2006), it also requires phonological encoding and representation (Coady and Evans 2008; Snowling et al. 1991).

Two recent meta-analyses suggest that NWR is also a promising method to identify language impairments in bilingual populations (Schwob et al. 2021; Ortiz 2021). Schwob et al. (2021) examine 46 studies on NWR in children (5 of them including only bilinguals, 9 including both monolinguals and bilinguals; age range: 2;0 to 9;4 years) qualitatively and 35 studies quantitatively. The meta-study of Ortiz (2021) is based on 13 studies



(6 of them including only bilinguals, 7 including both monolinguals and bilinguals; age range: 3;0 to 11;6 years). Both meta-analyses emphasize the high variability in the characteristics of the samples (such as the reference standard used, the types of bilinguals and the sample sizes), the stimuli (the syllable number and phonological complexity), and the scoring (the percentage of correct consonants and the whole item accuracy). With regard to the present study, the most important findings of both meta-analyses are that there are consistent effects of clinical status (TD vs. DLD) and that many of the NWR tests show good to very good diagnostic values for monolinguals and bilinguals. Ortiz (2021) reports larger effect sizes and better discrimination for language-independent compared to language-specific items, whereas Schwob et al. (2021) find no statistically relevant differences between language-independent and language-dependent items. In the meta-analysis, the term 'language independent' means that the items have as few language-specific properties as possible and are thus appropriate to test children speaking various languages. 'Language-specific' means that the nonwords closely correspond to the existing words of a particular language, for example by including existing morphemes, phoneme combinations, or stress patterns. Schwob et al. (2021) find no significant effects of age on diagnostic accuracy (i.e., the accuracy did not increase with age), but they point to the limited age range of the studies and argue that children aged 8 and older should be included more systematically in future studies. Both meta-analyses stress the need to use additional measures, such as sentence repetition, lexical tasks, narration, and parental questionnaires, for more reliable identification of DLD and emphasize that the results should be interpreted with some caution due to methodical differences and the small number of studies included.

The present study takes up some of these open issues by comparing monolingual and early second language learners (eL2) of German on their performance in the German LITMUS-QU-NWR (see below for more details). We focus on children aged 8 to 10, who have rarely been addressed so far (Schwob et al. 2021). The study also calculates the diagnostic accuracy, which has rarely been performed for this age group and any NWR. The results add further empirical evidence on the usability of the LITMUS-QU-NWR tasks to identify DLD in bilingual (and monolingual) acquisition.

### 1.1. The Two Types of LITMUS-NWR Tests

The results of the meta-studies (Schwob et al. 2021; Ortiz 2021) suggest that (most) NWR tasks can help to identify language impairments in bilinguals. This holds in particular if quasi-language-independent stimuli are used (Ortiz 2021).

The idea to design NWR tasks which distinguish between a language-independent and language-dependent part goes back to the COST Action IS0804 (www.bi-sli.org, accessed on 8 August 2022). In the COST Action, two major strategies for constructing nonwords were pursued (Chiat 2015; Ferré et al. 2015). The first type of NWR task, the so-called LITMUS Crosslinguistic Nonword Repetition task (LITMUS-CL-NWR), contains two main parts (see Chiat and Polišenská 2016 for more details). The first sub-part, the so-called cross-linguistic test (CLT), contains 16 items varying in length from two to five syllables (e.g., /lumi/, /mɑlitu/, /zipɑlidɑ/, /duligɑsumu/ in the British English version, see Chiat and Polišenská 2016). The items have a simple CV syllable structure. This sub-part taps into the phonological working memory. The second sub-part includes language-specific properties and is divided further into two parts. The so-called prosodically specific test (PST) uses the same items as the CLT but assigns word stress according to the rules of the specific language. The so-called language-specific test (LST) contains morphemes or phonotactics of a particular language and is more word-like (e.g., /rɪˈvaɪk/, /ˈfræʃək /, for British English). This sub-part taps into phonological working memory and language-specific knowledge (Chiat and Polišenská 2016; Boerma et al. 2015). The studies show good to excellent diagnostic accuracies for bilinguals (e.g., Boerma et al. 2015). The LITMUS-CL-NWR has been adapted to various languages (see bi-sli.org for an overview).

The second type of LITMUS-NWR is called the LITMUS Quasi-Universal Nonword Repetition task (LITMUS-QU-NWR). This type of NWR aims to tap into children's phono-

logical knowledge (Grimm and Schulz 2021; Grimm and Hübner forthcoming; Ferré et al. 2015). The rationale is that children with DLD have difficulties with phonological complexity (Marshall et al. 2003; Ferré et al. 2012). The pre-version has been developed for French and German and has been adapted to several other languages (see bi-sli.org for a list of languages). The LITMUS-QU-NWR comprises two parts: a so-called (quasi)language-independent part and a language-dependent part. In the language-independent part, phonological complexity is operationalized by using consonant clusters. The items of this part contain typologically widely attested vowels (/a/, /i/, /u/) and consonants (/p/, /k/, /f/, and a liquid) combined into one- to trisyllabic items of different syllable complexity (e.g., /ˈfluka/, /ˈklipafu/ for the German version). The language-dependent part adds a language-specific phonological structure, i.e., /s/ in the onset and coda position (e.g., /ˈʃpafika/, /ˈskiflapu/, /ˈsklifu/ in German; see Ferré et al. 2015 for modifications in French). Hence, in contrast to the LITMUS-CL-NWR, the term 'language-dependent' does not denote lexical similarity to the target language but refers to the language-specific phonological representation of /s/ in consonant clusters. Comprising two- to three-member clusters, the language-dependent part is inherently more complex than the language-independent part, which maximally contains two-member clusters. The German LITMUS-QU-NWR, which is the focus of this study, has two variants: a so-called long version (66 items, the pre-version) and a short version comprising 40 items. The short version is the result of a discriminant function analysis which was conducted to reduce the long version to items that best discriminate TD and DLD (see Grimm and Hübner forthcoming; Grimm and Schulz 2021 for more information). The material of the short version of the German LITMUS-QU-NWR will be described in more detail in the method section.

*1.2. Previous Results on the LITMUS-QU-NWR*

1.2.1. Group Comparisons

With regard to *group effects*, a number of studies indicate that the LITMUS-QU-NWR provides a reliable tool to identify DLD in bilingual and monolingual children. To date, children aged between 5 and 9 are most frequently represented in studies using the LITMUS-QU-NWR (dos Santos and Ferré 2016; de Almeida et al. 2017, 2019; Tuller et al. 2018; Abed Ibrahim and Fekete 2019; Abed Ibrahim and Hamann 2017; Hamann and Abed Ibrahim 2017; Chilla et al. 2021; Scherger 2020; Grimm and Schulz 2021; Ferré et al. 2015; Somberg 2020). The precise age range varies considerably between the studies: around age 5 (Grimm and Schulz 2021); between the ages of 5 and 6 (Wilkens et al. 2018); between the ages of 5 and 9 (Hamann and Abed Ibrahim 2017; dos Santos and Ferré 2016; Abed Ibrahim and Fekete 2019; Tuller et al. 2018); between the ages of 6 and 8 (Scherger 2020); or between the ages of 8 and 10 (Grimm and Hübner forthcoming). One further study examined the performance of refugees aged 7 to 11 years (Abed Ibrahim et al. 2020).

All these studies show the consistent effects of clinical status (i.e., TD outperformed DLD) but no negative (or positive) effects of bilingualism in TD children (i.e., monolingual vs. bilingual), independently of whether the initial classification was based on judgements by experts (Grimm and Hübner forthcoming), on independent language tests (Grimm and Schulz 2021; Abed Ibrahim and Fekete 2019), or a combination of tests and referral to a therapy (e.g., Somberg 2020; Scherger 2020; Tuller et al. 2018; dos Santos and Ferré 2016; Ferré et al. 2015). This stands in contrast to studies that report monolingual–bilingual differences in TD children for other NWR tests (e.g., Gutiérrez-Clellen and Simon-Cereijido 2010; Boerma et al. 2015). The majority of studies on the LITMUS-QU-NWR also find no significant differences between the monolingual and bilingual children with DLD (e.g., Grimm and Hübner forthcoming; de Almeida et al. 2017, 2019; Abed Ibrahim and Hamann 2017; Abed Ibrahim et al. 2020; Abed Ibrahim and Fekete 2019). In two studies, the Bi-DLD even outperformed the Mo-DLD (dos Santos and Ferré 2016; Ferré et al. 2015)[1]. These findings are consistent with the assumption that there are no cumulative negative effects of multilingualism and DLD (Paradis 2005, 2007; Armon-Lotem et al. 2015). Comparing the two parts, the effects are almost always stronger in the language-dependent

than in the language-independent part (Somberg 2020; Grimm and Hübner forthcoming; Grimm and Schulz 2021; Abed Ibrahim and Fekete 2019; Scherger 2020; dos Santos and Ferré 2016).

A few studies on the LITMUS-QU-NWR take additional factors, such as L1, age, age of onset, exposure, and SES, into account (Chilla et al. 2021; de Almeida et al. 2017; Tuller et al. 2018). The L1 does not significantly affect the performance in the LITMUS-QU-NWR (see Chilla et al. 2021 for a comparison of children with L1 Arabic, Turkish, and Portuguese). This finding is in line with the overall observation that, in early learners, the L1 effects are transient (Fennell and Tsui 2020) and characteristic of the early stages of phonological development (Kehoe 2015) and that differences in monolinguals disappear after "a couple of years" (Holm and Dodd 2006, p. 307). Within the bilingual groups, factors of language exposure or language use were not significantly related to the performance in the LITMUS-QU-NWR (Tuller et al. 2018; de Almeida et al. 2017; Chilla et al. 2021), but factors such as positive early development and, to a lesser extent, age could modulate differences within the bilingual group (see Tuller et al. 2018; de Almeida et al. 2017 for a discussion). Comparing simultaneous-bilingual (2L1, age of onset at birth until age 2) eL2 learners (age of onset between the ages of 2 and 4) and iL2 (age of onset after age 5) learners by their age of onset, Somberg (2020) finds no group differences. SES plays a minor role for the LITMUS-QU-NWR (Tuller et al. 2018); however, as pointed out by Tuller et al. (2018), small effects might have been undetected due to the limited sample size and limited range in age and exposure. Evidence from other NWRs suggests that chronological age and exposure can influence the performance (e.g., Sorenson Duncan and Paradis 2016; Thordardottir and Brandeker 2013), but this mostly confirms that SES is negligible in connection with nonword repetition (Chiat and Roy 2007; Boerma et al. 2015; Engel de Abreu et al. 2008; Chiat and Polišenská 2016; but see Meir and Armon-Lotem 2017).

Most studies on the LITMUS-QU-NWR examine the whole item accuracy (Abed Ibrahim and Fekete 2019; Somberg 2020; Scherger 2020; Tuller et al. 2018; de Almeida et al. 2017; Grimm and Schulz 2021). Whole item accuracy means that an item is scored as correct if the child's repetition corresponds to the target nonword in the number and order of phonemes, but the precise criteria for accuracy can differ across studies. Qualitative analyses have rarely been conducted for the LITMUS-QU-NWR. Among other findings, two studies provide further evidence that children with DLD struggle with phonological complexity such as internal codas and branching onsets (de Almeida et al. 2019; Ferré et al. 2015). Furthermore, Schallenberger (2021) finds that, depending on age and type of learner (monolingual, 2L1, eL2), different items of the German short version discriminate between TD and DLD.

Taken together, the quantitative and qualitative results so far suggest that the LITMUS-QU-NWR, particularly the language-dependent part, reliably discriminates between TD and DLD in bilingual (and monolingual) children. In addition, no monolingual–bilingual differences in TD learners are found, suggesting that the test does not penalize bilingual learners.

### 1.2.2. Diagnostic Accuracy

One crucial indicator of the usability of a test is its diagnostic accuracy. Diagnostic accuracy can be measured by sensitivity and specificity, by receiver operating characteristics (ROC), and by likelihood ratios, among others (Šimundić 2009). Sensitivity denotes the proportion of subjects within the pool of subjects who have the disorder and who are identified as having the disorder by the test (positive subjects). It indicates the potential of a test to detect subjects with the disorder (i.e., the percentage of correct positives). Specificity expresses the proportion of subjects that the test identifies as not having the disorder in relation to the total number of subjects without the disorder, i.e., the correct negatives. According to Friedman et al. (2020), perfect sensitivity and specificity are rare and more likely to occur in smaller sample sizes. Moreover, as pointed out in the literature (e.g., Armon-Lotem and Meir 2016; Plante and Vance 1995), high sensitivity rates typically go along with lower specificity rates. Plante and Vance (1994) propose that given

a sensitivity or specificity of less than 80%, a clinician cannot confidently use the test to identify children as TD or DLD. Given the tradeoff between sensitivity and specificity, they argue for lower specificity rates instead of lower sensitivity rates because subsequent tests will identify false positives and because the social consequences of false positives are less serious than those of false negatives (Plante and Vance 1995, p. 70). Based on these considerations, Plante and Vance (1995) consider specificity rates of 80% or higher as good and of 70% to 79% as fair.[2]

The optimal cutoff point to determine sensitivity and specificity can be calculated by a so-called receiver operating characteristic (ROC) curve analysis. In this analysis, the accuracy of the test is indicated by the area under the curve (AUC). The AUC value provides the overall accuracy of a test. An area of 1 indicates a perfect test; an area of 0.90 to 1 an excellent test; an area of 0.80 to 0.89 a good test; an area of 0.70 to 0.79 a fair test; and an area of 0.60 to 0.69 denotes a poor test. Values below 0.60 indicate an unacceptable test.

Another option to determine the diagnostic accuracy is the likelihood ratio (LR). The LR is calculated based on sensitivity and specificity. The LR for positive test results (LR+) indicates how much more likely the positive test result is to occur in subjects with the disorder compared to subjects without the disorder. LR+ is usually higher than 1 because is it more likely that the positive test result will occur in subjects with the disorder than in subjects without the disorder. LR+ is calculated according to the formula LR+ = sensitivity/(1-specificity). The LR for the negative test result (LR−) indicates how much less likely the negative test result is to occur in a subject with the disorder than in a subject without the disorder. It is calculated according to the formula LR− = (1-sensitivity)/specificity. Optimal tests have an LR+ of 10.0 or higher and an LR− of < 0.1, i.e., in these cases, the disorder is present or absent with high confidence. An LR+ of ≥ 3.0 and an LR− of ≤ 0.3 are 'suggestive'. An LR+ of < 3.0 and a LR− of > 0.3 indicate that the test does not discriminate between disorder and non-disorder.

To date, only a few studies have calculated the diagnostic accuracy for the LITMUS-QU-NWR as an index test (Somberg 2020; Tuller et al. 2018; Hamann and Abed Ibrahim 2017; Abed Ibrahim and Fekete 2019; dos Santos and Ferré 2016). These studies concentrated on children aged 5;6 to 8;11 and provided separate analyses for monolingual and bilingual children. The results demonstrate excellent to fair diagnostic values depending on the test part (language-independent, language-dependent, whole part) and group (monolingual, bilingual). In general, both the French and German long version (Tuller et al. 2018; Hamann and Abed Ibrahim 2017; Somberg 2020; dos Santos and Ferré 2016) and the German short version (Abed Ibrahim and Fekete 2019; Somberg 2020) discriminate between TD and DLD in monolingual and bilingual children with high confidence. A further consistent finding in both countries is that the diagnostic values are better for monolingual than for bilingual children (Tuller et al. 2018; Abed Ibrahim and Fekete 2019; dos Santos and Ferré 2016; de Almeida et al. 2017). This could be due to the locations of the recruitment of the monolinguals and bilinguals (dos Santos and Ferré 2016), different severities of the disorder (dos Santos and Ferré 2016), and the lack of appropriate tests for bilinguals as well as methodical decisions if children have to be classified as TD or DLD (de Almeida et al. 2017; Tuller et al. 2018; Abed Ibrahim and Fekete 2019).[3]

Out of the studies mentioned above, the study of Somberg (2020) and Grimm and Hübner (forthcoming) are of particular interest for the present research. Somberg (2020) conducts three analyses based on a bilingual population aged 5;5 to 9;0 (monolinguals are not included). In a first step, comparing the long and the short version, she finds that both versions discriminate between TD and DLD. For both versions (short/long), she receives the best diagnostic values if the two test parts, language-independent and language-dependent, are combined. In a second step, she splits the bilingual group according to their age of onset (AoO) to German into simultaneous-bilingual (2L1, AoO between 0 and 2 years), eL2 learners (AoO between 2 and 4 years), and late bilingual (lL2, AoO after 5 years) learners. The descriptive data suggest that eL2-TD learners reach lower percentages of correct repetitions and show more variation in the language-dependent

part compared to 2L1 learners. However, these differences are not statistically significant. Somberg (2020) finds the best diagnostic values for 2l1, eL2, and lL2 learners if both test parts are combined. Looking at the individual test parts, the language-dependent part better discriminates the 2L1 and lL2 groups, which is consistent with previous results. Surprisingly, diagnostic accuracy is better in the language-independent part if the eL2 group is considered. Unfortunately, as the study of Somberg (2020) does not include monolingual children, it remains an open question how eL2 learners perform in relation to monolingual children and if the diagnostic values are in general better for monolinguals, as reported in the literature (Tuller et al. 2018; Abed Ibrahim and Fekete 2019; dos Santos and Ferré 2016). The second study conducted by Grimm and Hübner (forthcoming) compares monolingual and bilingual children (*N* = 92) classified as TD or DLD by experts as a reference standard with regard to the performance in the short version of the test. The results confirm the findings of the younger age groups: there are no effects of language group, but clear effects of clinical status, and the effects are stronger in the language-dependent than in the language-independent part. However, the study does not analyze the diagnostic accuracy and does not distinguish between different types of bilingual learners.

### 1.3. Rationale of the Study and Research Questions

In sum, there is growing evidence that the LITMUS-QU-NWR is suitable to diagnose children aged 5 to 9. A group which has received little attention in relation to the LITMUS-QU-NWR comprises children aged 8 to 10 years (and also with regard to other NWRs, see Schwob et al. 2021). This lack of research is surprising in view of the fact that language impairments in bilinguals often remain undetected in the compulsory preschool screenings (Grimm and Schulz 2014; Voet Cornelli 2020). Furthermore, in bilinguals, the age of referral is often later than in monolinguals (Salameh et al. 2002). For example, the diagnosis of DLD in bilinguals often takes place between the ages of 6 and 8 in France (de Almeida et al. 2017, p. 334), despite the fact that compulsory screenings take place at age 5 (dos Santos and Ferré 2016). Among other factors (e.g., the lack of appropriate tools, see de Almeida et al. 2017; Tuller et al. 2018), one reason for the late referral is that compulsory language screenings come too early for many bilingual learners. In particular, successive-bilingual children have too little exposure to the L2 to allow a severe decision on whether the child is language-impaired or not. As a consequence, the decision and a referral to speech-language therapy is postponed (Voet Cornelli 2020; Salameh et al. 2002). This makes it likely that spoken language impairments in bilinguals will be detected during primary school, often in connection with written language difficulties.[4] These considerations justify the need to have diagnostic tools for children of primary school age. Nonword repetition provides a reliable method because it taps into both spoken and written language deficits (Bishop et al. 1996).

Despite the unambiguous need to have an instrument, there is a dramatic gap in the accessibility of reliable NWR tests for bilinguals (and monolinguals) of primary school age. For German, the Mottier test (Mottier 1951), which is composed of CV sequences of two to six syllables pronounced with equal duration and pitch, provides very good diagnostic values (AUC 0.96; sensitivity 90.1%; specificity 93.1%) for children between the ages of 7 and 10 with different types of language impairments (Kiese-Himmel and Nickisch 2015). However, the authors do not provide information on whether and how many bilinguals are included in the sample. Moreover, different types of bilingual children (Wild and Fleck 2013) are collapsed in the sample. A further NWR, standardized for children aged 5 to 12, is the subtest 'Kunstwörter' of the AGTB 5–12 (Hasselhorn et al. 2012). As with the Mottier test, this subtest uses CV sequences and hence also taps into the phonological working memory. Note that the subtest 'Kunstwörter' does not provide separate norms for monolingual and bilingual groups.[5]

The results so far suggest that the LITMUS-QU-NWR provides a reliable tool for identifying DLD in monolingual (Mo) and bilingual children; however, the available evidence suggests several gaps in the research. First, most studies on nonword repetition

do not keep 2L1 and eL2 learners apart from each other (but see Grimm and Schulz 2021; Somberg 2020; and Sorenson Duncan and Paradis 2016, who examined eL2 learners as a separate group). This covers the potential differences due to the later age of onset and limited exposure to the L2. Second, as pointed out by Schwob et al. (2021), there is little empirical research on NWR performance in children aged 8 to 10 and older. To our knowledge, no study so far examines the diagnostic accuracy of the LITMUS-QU-NWR for children of primary school age. The present study aims to fill these gaps by comparing the performances of Mo-TD, eL2-TD, Mo-DLD, and eL2-DLD children in both parts of the LITMUS-QU-NWR and analyzing the diagnostic accuracy. The two test parts are analyzed separately to see which part shows better diagnostic values for this age group. The following research questions are addressed:

(Q1) How do 8- to 10-year-old Mo-TD, eL2-TD, Mo-DLD, and eL2-DLD children perform in the language-independent and language-dependent part of the German LITMUS-QU-NWR if whole word accuracy is considered?

(Q2) Are the two parts suitable to diagnose DLD in monolingual and eL2 children aged 8 to 10 years?

Q1 addresses potential group differences in the two test parts. According to the aims of the test and based on the previous results, we expect effects of clinical status (TD vs. DLD) in both test parts. Regarding the language groups (Mo vs. eL2), we predict no differences in the language-independent part due to its quasi-universal construction and because the eL2 learners should be familiar with the types of clusters occurring in the language-independent items at the age of testing. No clear predictions can be made with regard to the language-dependent part. On the one hand, given that child L2 learners catch up rapidly to monolinguals in their phonological abilities (Fennell and Tsui 2020; Holm and Dodd 2006; Kehoe 2015), it is likely that the eL2 learners in our study have enough exposure to the phonological properties of the language-dependent items. If so, we do not expect effects of the language groups. On the other hand, given that the items are more complex and more (but not exclusively) specific to German, the eL2 learners might still struggle with the language-specific properties due to the limited exposure to German. In this case, we expect significant effects of language status in the language-dependent part. Considering that the literature reports little to no monolingual–bilingual differences for the 5- to 8-year-olds (Scherger 2020; dos Santos and Ferré 2016; Tuller et al. 2018; Abed Ibrahim and Fekete 2019) and no effects of the L1 (Chilla et al. 2021), we predict no difference between the eL2-TD and the Mo-TD groups aged 8 to 10.

Regarding Q2, we expect better diagnostic values for the language-dependent part compared to the language-independent part, in line with the literature (Somberg 2020; Grimm and Hübner forthcoming; Grimm and Schulz 2021; Abed Ibrahim and Fekete 2019; Scherger 2020; dos Santos and Ferré 2016). Furthermore, we predict a higher diagnostic accuracy for monolinguals, given the well-attested difficulties of diagnosing DLD in bilinguals (Tuller et al. 2018; dos Santos and Ferré 2016; Bedore and Peña 2008).

## 2. Materials and Methods

### 2.1. Recruitment

The participants—monolingual and eL2 learners aged 8;0 to 9;11 years—took part in project MILA (PI: P. Schulz, https://www.idea-frankfurt.eu/en/research/theme/individual-development/mila, accessed on 8 August 2022), conducted at the University of Frankfurt. Among other objectives, the project aimed at determining the indicators of DLD in bilingual children.

The monolinguals and eL2 learners were recruited between 2011 and 2012 in local schools and day-care centres and via SLTs. The recruitment took place as follows. In a first step, we sent an information letter to the institutions and asked for their participation. In the case of agreement, we asked the teachers and SLTs to distribute further information letters and consent forms to the parents of monolingual and eL2 learners aged 8 to 10. If the parents agreed to participate, we contacted them by telephone and conducted an interview

based on a questionnaire. The questionnaire was developed for project-internal purposes and collected information about the child's exposure to the L1 and L2, the language used at home and outside the family, the socio-economic background (operationalized as the mothers' years of schooling). The questionnaire also inquired about risk factors for early language development: attested hearing impairments and referral to a speech-language intervention, as well as spoken and written language impairments in the family (1st-grade relatives). The parents were also asked for attested cognitive, motor, and social conspicuities. In the eL2 group, the age of onset to the L2 was defined as the age of the first systematic contact with German. No restrictions were made with respect to the L1, i.e., the eL2 learners acquired different L1s (see below for more details). The project uses standardized tests to assess language abilities and the IQ. The language abilities in German are assessed via the TROG-D (Fox-Boyer 2016) in the case of the monolingual group, and via LiSe-DaZ (Schulz and Tracy 2011) in the case of the eL2 group (see below for more information). Based on the definition of DLD (Bishop 2017; Bishop et al. 2017), the IQ provides no exclusionary criterion.

### 2.2. Inclusionary Criteria

Children were included in the study if they showed no history of hearing impairment, age-appropriate motor development, and typical social or emotional development according to parental information (questionnaire).

### 2.3. Classification as TD or DLD

The classification as TD vs. DLD (=reference standard) was based on (a) the referral to a speech-language intervention due to an oral language deficit and (b) on the performance in a language test (Bossuyt et al. 2015; Dollaghan and Horner 2011). Children were classified as TD if they were never referred to speech-language therapy according to parental information and if they performed age-appropriately in the respective language test. Likewise, children were classified as DLD if they were referred to speech-language therapy and if they performed T < 40 in the language test. In the monolingual group, there is a perfect match of referral and result in the TROG-D, i.e., no over- or underdiagnosis. In the eL2 group, out of the original sample of $n = 36$ eL2 learners, the referral and test results match in 33/36 (91.7%) children. The three remaining children were all underdiagnosed (i.e., not referred despite a poor test result). These three children were excluded from the analysis, resulting in $n = 33$ eL2 learners.

As stated above, different language tests were used for the monolingual and the eL2 groups. In the monolingual group, we used the TROG-D (Fox-Boyer 2016), a standardized language test assessing sentence comprehension. The test provides norms for monolinguals aged 3;0 to 10;11 years and only has to be administered to monolingual children. Monolingual children were classified as DLD if they scored T < 40 in the TROG-D and if they were referred to speech-language therapy.

For the eL2 group aged 8 to 10, no comparable norm-referenced language test is available. We chose LiSe-DaZ, a standardized language test which has been constructed with a particular focus on eL2 learners, for two reasons. First, due to the focus on eL2 learners, LiSe-DaZ is culturally less biased than tests developed for monolingual children (e.g., the TROG-D). Second, LiSe-DaZ provides norms for eL2 learners between 3;6 and 7;11 years. We adopted the norms of the oldest age group and consider it meaningful for DLD if older eL2 children scored T < 40 in two or more subtests of LiSe-DaZ and if they were referred to speech-language therapy. Given the range of L1s, no testing in the L1 was possible, but the onset of the single-word and the multiword stage was considered as additional confirmation (see below for more information).

### 2.4. Participants

Table 1 shows the sample characteristics. All the monolingual children, TD or DLD, were born in Germany. German was the only language used by the family and was the

only language the child had acquired at the time of testing. The Mo-TD group comprised 13 girls and 14 boys and the Mo-DLD group 3 girls and 6 boys.

**Table 1.** Participant characteristics.

|  | Mo-TD (*n* = 27) | | Mo-DLD (*n* = 9) | | eL2-TD (*n* = 24) | | eL2-DLD (*n* = 9) | |
|---|---|---|---|---|---|---|---|---|
|  | *M* | *SD* | *M* | *SD* | *M* | *SD* | *M* | *SD* |
| Age | 113.0 | 10.1 | 110.3 | 8.2 | 110.1 | 7.9 | 112.8 | 8.2 |
| AoO | - | | - | | 36.8 | 7.4 | 39.4 | 9.6 |
| LoE | - | | - | | 73.3 | 12.2 | 73.1 | 11.1 |
| SES | 12.6 | 1.0 | 9.5 | 1.5 | 11.6 | 4.6 | 9.6 | 0.7 |

AoO: age of onset to German, LoE: length of exposure to German. Ages are given in months; SES in years.

The eL2 children, TD and DLD, are successive learners of German who started to acquire German between the ages of 2;0 and 3;11. All the children were born in Germany, except one eL2-DLD child, who was born in Kazakhstan. At the time of testing, all the eL2 children had attended German primary schools for at least two years and attended kindergartens in Germany before entering primary school. The eL2-TD children acquired 12 different languages at home: Hindi/Urdu (*n* = 5), Turkish (*n* = 4), Russian (*n* = 2), Japanese (*n* = 2), Serbian (*n* = 2), Arabic (*n* = 2), Tamil (*n* = 2), Italian (*n* = 1), Albanian (*n* = 1), Bangla (*n* = 1), and French (*n* = 1). The eL2-DLD children acquired 5 different languages at home: Arabic (*n* = 2), Serbian/Croatian (*n* = 2), Urdu (*n* = 1), Russian (*n* = 1), Pashto (*n* = 1). The eL2-TD group contained 15 girls and 9 boys, the eL2-DLD group 3 girls and 6 boys.

Shapiro–Wilk tests show that age, age of onset, length of exposure, and SES are not normally distributed (age: $W = 0.955$, $p = 0.025$; AoO: $W = 0.777$, $p < 0.001$; LoE: $W = 0.924$, $p = 0.001$; SES: $W = 0.752$, $p < 0.001$). Based on this outcome, non-parametric tests (Mann–Whitney U tests) are used to evaluate possible group differences. The comparisons show no significant age differences, either for the Mo-TD vs. Mo-DLD ($U = 99.5$, $z = -0.805$, $p = 4.28$) or the Mo-TD vs. eL2-TD ($U = 266.5$, $z = -1.087$, $p = 0.277$), the eL2-TD vs. eL2-DLD ($U = 88.5$, $z = -0.791$, $p = 0.437$), and the Mo-DLD vs. eL2-DLD groups ($U = 32.0$, $z = -0.756$, $p = 0.489$). The eL2 groups do not significantly differ in the age of onset of German ($U = 81.0$, $z = -1.094$, $p = 0.290$) and the exposure to German ($U = 108.0$, $z = 0.000$, $p = 1.0$). Regarding SES (measured in the mothers' length of schooling), there are significant differences between the Mo-TD and the eL2-TD group ($U = 8.5$, $z = -5.959$, $p < 0.001$), the eL2-TD and the eL2-DLD groups ($U = 44.0$, $z = -2.236$, $p = 0.03$), and the Mo-DLD group, which has a significantly lower SES compared to the Mo-TD group ($U = 14.0$, $z = -3.849$, $p < 0.001$). SES does not differ between the Mo-DLD and eL2-DLD groups ($U = 21.0$, $z = -1.242$, $p = 0.279$). Due to these differences, SES is considered in the statistical model.

Given that tests are not available for all L1s of the eL2 learners, the diagnosis is confirmed by two risk factors of early development: the age of the production of the first words and the age of the emergence of multiword utterances (Table 2). These factors can support the diagnosis if no assessment in the L1 is possible (Boerma and Blom 2017; Tuller 2015). In our eL2 group, the age of onset to German is after two years of age as per the definition and is factually around age 3 (see Table 1). This implies that the first words and the first word combinations should be produced in the L1. In that sense, the risk factors provide information about the development in the L1.

**Table 2.** Age at the emergence of first words and multiword utterances, in months.

| | Mo-TD (*n* = 27) | | Mo-DLD (*n* = 9) | | eL2-TD (*n* = 24) | | eL2-DLD (*n* = 9) | |
|---|---|---|---|---|---|---|---|---|
| | *M* | *SD* | *M* | *SD* | *M* | *SD* | *M* | *SD* |
| First words | 10.9 | 3.1 | 18.5 | 8.9 | 13.2 | 4.8 | 11.7 | 2.9 |
| Word combinations | 16.8 | 3.9 | 32.4 | 13.9 | 18.4 | 4.6 | 35.2 | 15.2 |

Following Grimm and Schulz (2014), we considered a child as delayed if the first words emerged after 18 months or if s/he entered the multiword stage later than 24 months of age. Table 3 depicts the distribution of the two risk factors within the groups. Shapiro–Wilk tests show that the two variables are not normally distributed (first words: $W = 0.851$, $p < 0.00$; word combinations: $W = 0.769$, $p < 0.001$). Subsequent statistical comparisons confirm the descriptive data. Group comparisons (Mann–Whitney U tests) reveal no significant differences between the Mo-TD and the eL2-TD ($U = 212.5$, $z = -1.754$, $p = 0.079$), between the eL2-TD and the eL2-DLD ($U = 70.0$, $z = -0.527$, $p = 0.631$), and between the Mo-DLD and the eL2-DLD groups ($U = 13.5$, $z = -1.695$, $p = 0.094$) regarding the production of the first words. The Mo-DLD produced their first words at a significantly later age than the Mo-TD group ($U = 44.5$, $z = -2.433$, $p = 0.013$). The Mo-TD produced multiword utterances at a significantly earlier age than the Mo-DLD children ($U = 22.0$, $z = -3.329$, $p < 0.001$); the same holds for the eL2-TD compared to the eL2-DLD learners ($U = 25.5$, $z = -3.324$, $p < 0.001$). No differences regarding the age at the production of multiword utterances emerged between the Mo-TD and the eL2-TD ($U = 229.5$, $z = -1.240$, $p = 0.215$) and between the Mo-DLD and the eL2-DLD groups ($U = 31.5$, $z = -0.437$, $p = 0.673$). Taken together, the qualitative and quantitative results indicate that the TD groups are hardly affected by the risk factors and that a majority of the children diagnosed as DLD show late development in at least one of the factors (mostly the emergence of multiword utterances).

**Table 3.** Number of children affected by the risk factors per group.

| | Mo-TD (*n* = 27) | Mo-DLD (*n* = 9) | eL2-TD (*n* = 24) | eL2-DLD (*n* = 9) |
|---|---|---|---|---|
| No risk | 26 | 3 | 21 | 3 |
| First words | 0 | 0 | 0 | 0 |
| Word combinations | 0 | 2 | 0 | 6 |
| Both factors | 0 | 3 | 2 | 0 |
| Missing data | 1 | 1 | 1 | 0 |

*2.5. Material*

The children were tested with the short version of the German LITMUS-QU-NWR. Like the long version, the short version of the LITMUS-QU-NWR contains a (quasi-)language-independent and a language-dependent part. In both parts, a trochaic 'CVCV shape forms the base and is varied systematically in phonological complexity. In the language-independent part, the 'CVCV shape is expanded by one or more consonants and/or by an additional syllable. The nonwords are composed of the vowels /a/, /i/, /u/ and the consonants /p/, /k/, /f/, and /l/. The language-dependent (language-dependent) items are constructed according to the same principles as the language-independent part plus /s/ or /ʃ/ in the word-initial and word-final positions (e.g., 'sCCVCV; 'CCVCVCs). The /s/ or /ʃ/ in the onset or coda position is chosen because its representation varies cross-linguistically (see Grimm and Schulz 2021; Grimm and Hübner forthcoming for more details). Each part consists of 20 items (see Appendix A for a list of items).

To test basic auditory skills, a pre-test on auditory discrimination precedes the NWR. The auditive or auditory discrimination task includes 24 bisyllabic nonword pairs (12 similar and 12 distinct pairs of nonwords; all stressed on the initial syllable) and focuses on the nasals and liquids in the word-initial (e.g., /luba/-/ruba/, /niwa/-/miwa/) and

word-medial positions (e.g., /sonə/-/somə/, /banu/-/bamu/). The nasals and liquids are chosen because they are particularly difficult to discriminate by hearing-disabled persons (Raphael 2008). Good performances in auditory discrimination are seen as evidence of age-appropriate hearing abilities at the time of testing (note that children with known hearing difficulties were initially excluded from participation; see inclusionary criteria). The children were included in this study if they responded correctly to at least 18/24 (75%) nonword pairs. No child was excluded due to poor auditory discrimination.

### 2.6. Procedure

Testing took place individually in a quiet room at the child's school. The items were presented via a laptop and headphones. To control the effects of the order of the items, four lists of items were created by a pseudo-randomization and evenly distributed across the participants.

The children are told that they will hear unknown and funny words and that they should listen carefully and repeat these words. The task starts with two training items. In the main test, the items are repeated once if necessary but excluded from the analysis. The sessions are recorded by a voice recorder and a highly sensitive microphone for later transcription.

### 2.7. Transcription and Data Coding

The children's productions were transcribed orthographically by the author or a trained student assistant who was not the investigator. Orthographic transcriptions are used because we do not aim to go into phonetic detail and because the phonological processes we count (omissions, additions, metatheses, and substitutions) are analysable under orthographic transcriptions.

The data coding is conducted in MS Excel. A repetition is considered correct if the consonants and their ordering correspond to the target form. Vowel errors are very infrequent and not further considered. Changes in the voicing of consonants (/ˈpifakʊp/ → [ˈbifakʊp]) are not considered errors (dos Santos and Ferré 2016). Likewise, replacements of /ʃ/ by [s] or the interdental realization of /s/ are not considered errors since the sounds occur in clusters, where the opposition is not phonemic in German. Null reactions and repeated items are coded as missing data. Missing data occurred in 16/2761 trials, i.e., 0.6% of the data.

### 2.8. Data Analysis

The analysis considers the whole word accuracy, given as the percentage of correctly repeated items. Descriptive statistics and the ROC analysis were conducted in SPSS 27. The repetition accuracy was analysed with mixed-effect models using R (R Development Core Team 2012) and the R packages lme4 (Bates et al. 2015) and languageR (Baayen 2008). A mixed-effect logistic regression analysis with the function glmer was conducted. The dependent variable is whether participants repeat the whole word correctly or incorrectly. The model contains clinical status (TD, DLD), language group (Mo, eL2), their two-way interaction and SES as fixed factors. SES is included in the model because there are significant group differences. We added participants and items as our random intercepts (formula: accuracy ~ clinic status * language group + SES + (1 | subject) + (1 | item), family = binomial) (Barr et al. 2013; Jaeger 2008). The model was run once for the language independent part and once for the language-dependent part.

## 3. Results

### 3.1. RQ 1: Group Comparisons

The first research question addresses potential group differences. Table 4 presents the percentages of correctly produced items. The results show that all the groups struggle more with the language-dependent part than with the language-independent part: the number of correctly repeated items is lower in the language-dependent part than in the

language-independent part. Descriptively, the eL2 learners show lower performances than the monolingual peers in the language-dependent part, which suggests that eL2 learners have more difficulties producing the language-dependent items in general. The TD-DLD-differences are descriptively more pronounced in the LD compared to the LI part (monolingual: $\Delta_{LI} = 14.1\%$; $\Delta_{LD} = 22.2\%$; eL2: $\Delta_{LI} = 17.1\%$; $\Delta_{LD} = 25.2\%$). The SDs are high, which indicates overlapping performances of the TD and DLD children in both parts.

**Table 4.** Mean percentage of correctly repeated items and SD by group for the language independent (LI) and language-dependent (LD) part.

| | | LI Part | | LD Part | |
|---|---|---|---|---|---|
| | *n* | *% Correct* | *SD* | *% Correct* | *SD* |
| Mo-TD | 27 | 93.5 | 5.5 | 85.0 | 8.9 |
| Mo-DLD | 9 | 79.4 | 12.4 | 62.8 | 8.7 |
| eL2-TD | 24 | 91.5 | 9.2 | 78.5 | 11.3 |
| eL2-DLD | 9 | 74.4 | 12.6 | 53.3 | 12.9 |

For the language-independent part, we expect to find a significant effect of clinical status but no effect of the language group and no interaction of the two factors. Following previous studies using nonword repetition, SES is not expected to affect performance. The statistical results confirm these predictions (Table 5).

**Table 5.** Language-independent part: outcome of the mixed-effect logistic regression analysis.

| | Estimate | *SE* | *z* | *p* |
|---|---|---|---|---|
| Clinical status | 2.048 | 0.446 | 4.592 | $4.39 \times 10^{-6}$ *** |
| Language group | 0.448 | 0.485 | 0.922 | 0.356 |
| Clinical*language | −0.047 | 0.616 | −0.077 | 0.938 |
| SES | −0.015 | 0.048 | −0.311 | 0.756 |

*** $p < 0.001$.

Likewise, in the language-dependent part we expect to find an effect of clinical status but no effect of the language group due to the exposure to German of more than 6 years ($M = 73$ months) in the eL2 group. We also expect to find no effect of SES. The data confirm these predictions (Table 6).[6]

**Table 6.** Language-dependent part: outcome of the mixed-effect logistic regression analysis.

| | Estimate | *SE* | *z* | *p* |
|---|---|---|---|---|
| Clinical status | 1.901 | 0.367 | 5.181 | $2.2 \times 10^{-7}$ *** |
| Language group | 0.745 | 0.421 | 1.771 | 0.077 |
| Clinical*language | −0.189 | 0.508 | −0.372 | 0.709 |
| SES | 0.056 | 0.040 | 1.381 | 0.167 |

*** $p < 0.001$.

### 3.2. RQ 2: Diagnostic Accuracy

A receiver operating characteristic (ROC) curve analysis is conducted on the percentage of correctly repeated items in order to assess the optimal cutoff scores and the diagnostic accuracy of the language-independent and language-dependent parts separately. Likelihood ratios are calculated to obtain additional information on the diagnostic accuracy of the LITMUS-QU-NWR. Figure 1 presents the ROC curves for the monolingual group and Figure 2 for the eL2 group. The AUC values, cutoff score points, and levels of sensitivity, specificity, overall accuracy, and likelihood ratios for the language-independent and the language-dependent parts are shown in Table 7 for both groups.

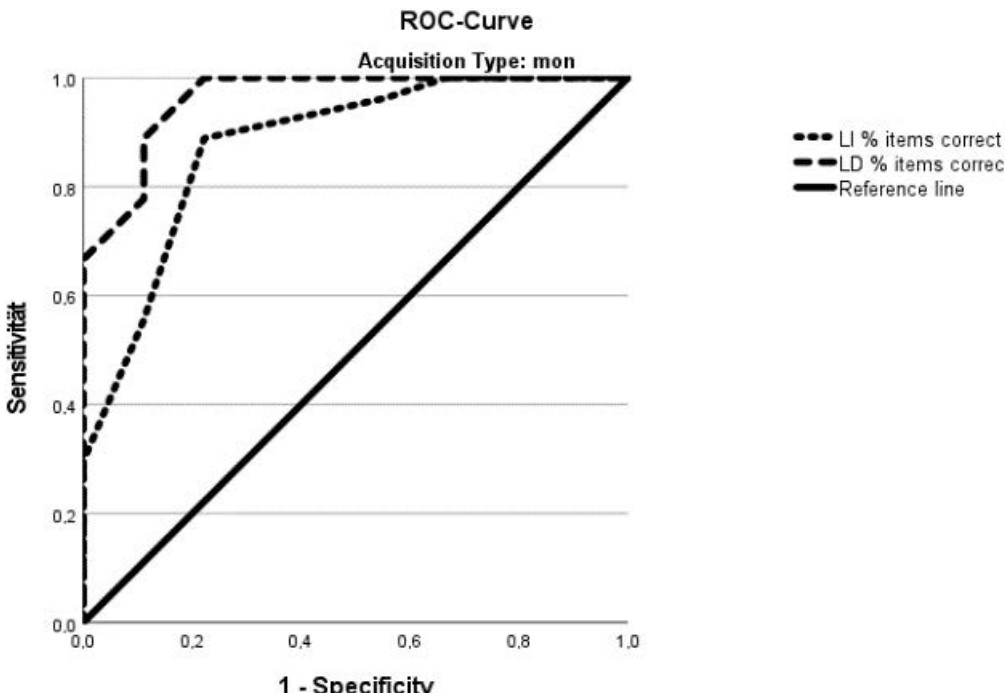

**Figure 1.** ROC curves for the language-independent (LI) and the language-dependent (LD) part for the monolingual group.

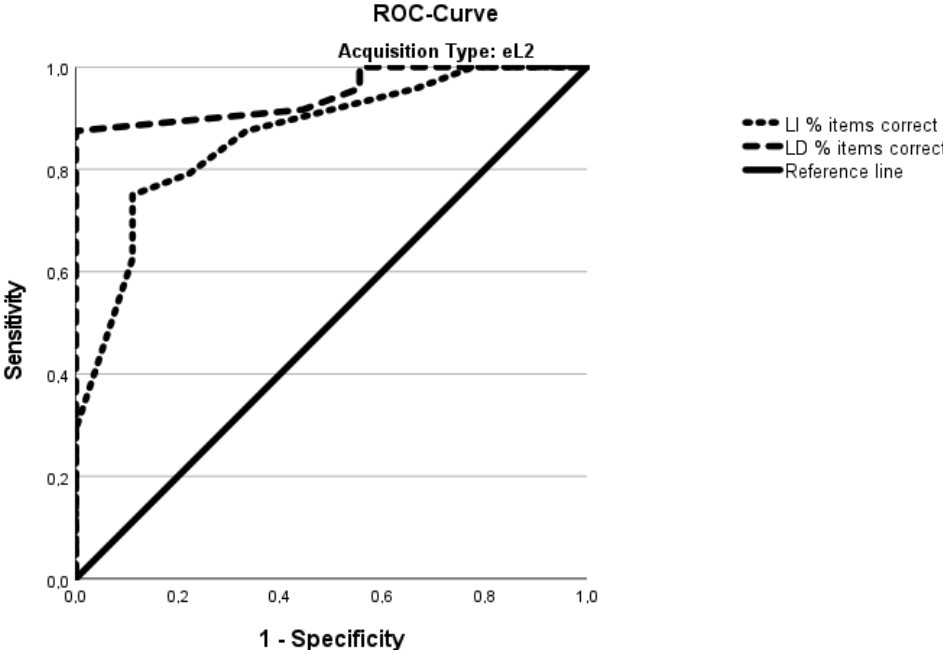

**Figure 2.** ROC curves for the language-independent (LI) and the language-dependent (LD) part for the eL2 group.

**Table 7.** Diagnostic metrics for the language-independent (LI) and the language-dependent part (LD) by acquisition type.

|  | Mo (*n* = 36) | | eL2 (*n* = 36) | |
| --- | --- | --- | --- | --- |
|  | **LI Part** | **LD Part** | **LI Part** | **LD Part** |
| AUC | 0.879 * | 0.963 *** | 0.866 ** | 0.946 *** |
| Cutoff (%) | 87.5 | 72.5 | 82.5 | 67.5 |
| Sensitivity (% correct DLD) | 88.9 | 88.9 | 79.2 | 87.5 |
| Specificity (% correct TD) | 77.8 | 88.9 | 77.8 | 100 |
| LR+ | 4.0 | 8.1 | 3.6 | n.a |
| LR− | 0.2 | 0.1 | 0.3 | 0.0 |

*** $p < 0.001$; ** $p < 0.01$; * $p < 0.05$.

The results indicate excellent to good AUC values for the sample of this study, depending on the part of the NWR. The diagnostic values are better in the language-dependent compared to the language-independent part. Under the proposed cutoffs we obtain better sensitivities than specificities in the language-independent part. For the monolingual group, sensitivity equals specificity in the language-dependent part. In the eL2 group, the specificity is perfect, indicating that all TD children are identified as TD by the index test. According to Plante and Vance (1994), the sensitivity is good to fair and the specificity ranges from perfect to fair. For the monolingual group, the likelihood ratios indicate good values for the language-dependent part and suggestive values for the language-independent part. In the eL2 group, the language-independent part also shows suggestive values. LR+ cannot be calculated for the language-dependent part under the formula LR+ = sensitivity/(100 − specificity) because the denominator becomes zero if the specificity is 100%.

## 4. Discussion

The present study explores whether the short version of the German LITMUS-QU-NWR can be used to disentangle DLD 8- to 10-year-old children classified as (non-)DLD by referral to SLT and a language test. This particular age group has rarely been investigated in connection with the LITMUS NWR tools in any language. Furthermore, this study expands prior research by comparing monolingual children and eL2 learners of German. In doing so, the study provides new evidence that eL2 learners aged 8 to 10 years do not differ from monolingual children with regard to particular phonological abilities (here: production of consonant clusters). This is important because most studies on LITMUS-NWR tasks conflate 2L1 and eL2 learners, who are inherently highly heterogeneous. Two research questions are addressed: How do 8- to 10-year-old Mo-TD, eL2-TD, Mo-DLD, and eL2-DLD children perform in the language-independent and language-dependent part of the German LITMUS-QU-NWR if whole item accuracy is considered? Are the two parts suitable to diagnose DLD in monolingual and eL2 children aged 8 to 10 years?

To answer RQ1, we analyzed the group differences in the rates of correctly repeated items in the language-independent and language-dependent part, based on whole item accuracy. We found no effects of language group, i.e., no differences between Mo-TD and eL2-TD and between Mo-DLD and eL2-DLD, but significant effects of clinical status, i.e., between Mo-TD and Mo-DLD and between eL2-TD and eL2-DLD. These results strongly suggest that in principle the LITMUS-QU-NWR discriminates DLD and TD in monolinguals and eL2 learners on a group level. In order to determine the diagnostic accuracy of the two parts, we conducted ROC analyses and likelihood ratios (RQ2). The measures of diagnostic accuracy indicate very good diagnostic values for the monolingual and for the eL2 group, with better diagnostic values for the language-dependent compared to the language-independent part. The results and their implications will be discussed below.

### 4.1. RQ 1: Group Differences

The overall rationale of the LITMUS-QU-NWR is to design a test that assesses phonological complexity, that does not penalize bilingual learners because of their shorter exposure to the test language, and that disentangles TD from DLD in monolingual and in bilingual populations (dos Santos and Ferré 2016; Ferré et al. 2015). The aim of the language-independent part is to identify DLD in bilinguals and monolinguals (together with other instruments) based on the cross-linguistically well-attested criteria of phonological complexity. Due to the (quasi-)universal properties of the language-independent part, we expect that the differences between the language groups (Mo-TD vs. eL2-TD) are negligible in the age group we examine but that the TD will outperform the respective DLD peers. The language-dependent part seeks to examine children's language-specific phonological knowledge. Given that bilingual children, and in particular eL2 learners, have less exposure to these language-specific properties than their monolingual peers, differences between Mo-TD and eL2-TD are expected in younger age groups, in children who have little exposure to the L2 and if the tested properties do not occur in the L1. These monolingual–bilingual differences will disappear after a certain amount of exposure to the L2 (Holm and Dodd 2006; Fennell and Tsui 2020; Kehoe 2015). Moreover, the evidence so far suggests that exposure and SES play only a small (Tuller et al. 2018), or no, role (Abed Ibrahim and Fekete 2019) for bilingual German-speaking children between the ages of 5 and 9. In addition, no effect of the L1 (Chilla et al. 2021) is found for the LITMUS-QU-NWR-German. Based on these findings and considering the fact that the participants of the present study have even more experience with L2 German than the participants in the previous studies, we predict no statistical differences between Mo-TD and eL2-TD children in the language-dependent part. However, a significant effect of clinical status is expected. Both predictions are confirmed by the statistical results. Consistent with our expectations, we find no differences between the TD groups but significant differences between the TD and the DLD groups independently of which test part is used. These results suggest that the test is unbiased for both monolingual and eL2 learners of this age group and that the test does not disadvantage eL2 learners because of their shorter exposure to German.

Our findings expand the previous results of the LITMUS-QU-NWR and other NWR tasks in several respects. First of all, the results provide further evidence that the LITMUS-QU-NWR discriminates TD and DLD in the group of children aged 8 to 10 (Grimm and Hübner forthcoming). This is important given the later age of diagnosis in bilingual learners (de Almeida et al. 2017; Salameh et al. 2002; Voet Cornelli 2020). Second, in line with previous studies, the descriptive results indicate stronger TD-DLD differences in the language-dependent part than in the language-independent part (monolingual: $\Delta_{LI} = 14.1\%$; $\Delta_{LD} = 22.2\%$; eL2: $\Delta_{LI} = 17.1\%$; $\Delta_{LD} = 25.2\%$). This is in line with previous findings (Somberg 2020; Grimm and Hübner forthcoming; Grimm and Schulz 2021; Abed Ibrahim and Fekete 2019; Scherger 2020; dos Santos and Ferré 2016). This finding is not surprising in view of the higher structural and articulatory complexity of the language-dependent part and in view of the fact that children and adolescents with DLD experience difficulties in the processing of linguistic complexity (Marshall et al. 2003; Ferré et al. 2015; de Almeida et al. 2019; Ferré et al. 2012). The results provide further empirical evidence that children with DLD still struggle with phonological complexity (here operationalized as clusters) at school age. Finally, in line with Tuller et al. (2018), we find no effects of SES, indicating that the LITMUS-QU-NWR is not only culturally but also socially less biased.

### 4.2. RQ 2: Diagnostic Accuracy

This part of the study aims to examine the diagnostic accuracy of the LITMUS-QU-NWR for children aged 8 to 10. The aim is to see how well the test disentangles TD and DLD in monolingual children and in eL2 learners of German. In order to examine the efficiency of the NWR task, we conduct ROC analyses and likelihood ratios separately for the monolingual and the eL2 group and the two test parts.

In the ROC analyses, we find good to excellent AUC values for the monolingual group (language-independent 0.879; language-dependent 0.963) and the eL2 group (language-independent 0.866; language-dependent 0.946). With regard to the cutoffs, several previous studies on the LITMUS-QU-NWR calculate the diagnostic accuracy for the whole test and apply the same cutoff for monolingual and bilingual children (Tuller et al. 2018; Abed Ibrahim and Hamann 2017; dos Santos and Ferré 2016). Due to the increasing evidence that the language-dependent part better discriminates between TD and DLD than the language-independent part (Somberg 2020; Grimm and Hübner forthcoming; Grimm and Schulz 2021; Abed Ibrahim and Fekete 2019; Scherger 2020; dos Santos and Ferré 2016), this study takes a closer look at the diagnostic usability of the two test parts. The ROC analyses result in different cutoffs for each part and group (language-independent part: Mo 87.5%; eL2 82.5%; language-dependent part: Mo 72.5%, eL2 67.5%). This finding is in line with Abed Ibrahim and Fekete (2019), who report different cutoffs for monolingual and for bilingual 5- to 8-year-old children in both parts of the test as well. Grimm and Schulz (2021) calculated different cutoffs for 5-year-old 2L1 and eL2 learners in the language-dependent part (2L1: 37.5%; eL2: 31.2%) but not in the language-independent part. In contrast, comparing 2L1 and eL2 learners aged 5 to 8;11, Somberg (2020) finds the same cutoff for 2L1 and eL2 learners in the language-independent part (59.4%) and in the language-dependent part (46.9%). Note that all these studies differ with regard to the sample characteristics (i.e., age, type of learner, and identification as DLD), leaving open the question regarding the conditions under which the same cutoff can be used for all types of bilinguals. Comparing monolinguals and eL2 learners, the present study suggests that separate cutoffs should be used even at age 8 to 10 and after approximately 6 years of exposure ($M \sim 73$ months) to the L2 German.

Under the calculated cutoffs, we mostly obtain better sensitivity than specificity values. Sensitivity and specificity are identical for the monolingual group in the language-dependent part. In the monolingual group, the sensitivity is good in both test parts (language-independent and language-dependent: 88.9%). The specificity values are good in the language-dependent part (88.9%) but only fair in the language-independent part (77.8%). In the eL2 group, a good sensitivity is found in the language-dependent part (87.5%) but only a fair sensitivity in the language-independent part (77.8%). The specificity is fair in the language-independent part (77.8%) and perfect (100%) in the language-dependent part. The best likelihood ratios are achieved in the language-dependent part for the monolingual group (LR+ 8.1; LR− 0.1); the values for the language-independent part are only suggestive (LR+ 4.0; LR− 0.2). The language-independent part is also suggestive for the eL2 learners (LR+ 3.6, LR− 0.3). Due to the specificity of 100%, the LR cannot be computed for the language-dependent part. These diagnostic values resemble the excellent to good values reported for younger monolingual and bilingual populations in Germany but differ from the French data which convey lower diagnostic accuracies in the bilingual samples (Tuller et al. 2018).

Why do we find similar diagnostic accuracies in the monolingual and eL2 groups? First of all, the location of recruitment can influence the diagnostic accuracy (Tuller et al. 2018; de Almeida et al. 2017; dos Santos and Ferré 2016). As argued in dos Santos and Ferré (2016, p. 11), the Mo-DLD children in their study are recruited in hospitals and are more severely impaired compared to the Bi-DLD children who are recruited via SLTs. In our study, the Mo-DLD and eL2-DLD children are recruited in the same institutions. Hence, effects of the recruitment on the accuracy of the initial classification are less likely. Secondly, at age 8 to 10, the gap between children with DLD and TD children becomes bigger and language deficits become more pronounced. This makes it likely that the referral and medical test will match. In fact, we observe a perfect match between referral and the result of the TROG-D in the monolingual group and a high agreement between referral and the result of the LiSe-DaZ (91.3%) in the eL2 group.

At first glance, the better diagnostic values found in the language-dependent part seem to contradict the results of the meta-studies that report better results for the language-

independent items (Ortiz 2021) or no difference between the test parts (Schwob et al. 2021). This has to do with the construction of the tests and the small number of studies available so far. The majority of NWR tasks included in the meta-studies of Schwob et al. (2021) and Ortiz (2021) use CV sequences or less word-like items in the language-independent part and word-like items (in relation to a particular language) in the language-dependent part. The TD-DLD differences in bilinguals can be less pronounced because even typically developing bilinguals often struggle with nonwords that are very similar to existing words in a particular language (Gutiérrez-Clellen and Simon-Cereijido 2010; Boerma et al. 2015).

The LITMUS-QU-NWR, however, follows different principles of construction: the language-independent part is composed of cross-linguistically widely attested sounds, and the language-dependent part adds language-specific phonological complexity. Compared to other NWRs, the language-dependent part of the LITMUS-QU-NWR is still less word-like and shows little similarity to words in German. Hence, given that both test parts involve complex phonological structures and that children with DLD struggle with the linguistic complexity, we expect that both parts discriminate between TD and DLD. The more pronounced effects between TD vs. DLD children in the language-dependent part presumably have to do with its higher inherent complexity in combination with the language-specific clusters. If the difficulties repeating nonwords increase with the phonological complexity in monolingual and in bilingual children with DLD, the language-dependent part should be more valuable for diagnostic purposes, at least in older children.[7]

In sum, the present study demonstrated that the language-dependent part of the LITMUS-QU-NWR alone identifies children with DLD aged 8 to 10 with high confidence. Hence, the test can help to identify children with DLD in this age group. This is an important result given the lack of NWR tasks for this age group. However, as emphasized in the literature, the identification of DLD should not be based on a single test (Armon-Lotem and Meir 2016; Tuller et al. 2015, 2018; Boerma and Blom 2017; Schwob et al. 2021).

## 5. Limitations and Further Directions

The present study provides important insights into the usability of the LITMUS-QU-NWR as an index test; however, the results should be interpreted in the light of some limitations. First, future research should examine in more detail how linguistic and non-linguistic factors such as SES, language exposure, and the structure of the L1 influence the performance. In our study, we find no effects of SES (measured as the mothers' length of schooling), in line with research on other NWRs (e.g., Schwob et al. 2021; Engel de Abreu et al. 2008). Given that the eL2-TD and eL2-DLD groups are matched for exposure and age of onset, the measures of language exposure cannot account for the observed differences in the eL2 group (Tuller et al. 2018; de Almeida et al. 2017). However, in our and the previous studies, the range in these variables is rather limited, minimizing the chance to find statistical effects. Furthermore, the present study also ignores the potential effects of the L1. Although systematic comparisons uncover no L1 effects (Chilla et al. 2021), L1 effects are possible, particularly if the children are younger and have had only short exposure to the test language. Our sample, however, had substantial exposure to L2 German at the time of testing. Therefore, we consider it unlikely that the L1 explains much of the TD-DLD differences. A further limitation is that the diagnostic accuracy is potentially overestimated. As pointed out by Friedman et al. (2020), smaller sample sizes—as in our study—increase the likelihood of obtaining excellent diagnostic values. In addition, due to the lack of normed tests for the age group we examine, the diagnosis of (non-)DLD in the eL2 group is based on the norms for younger eL2 children, and thus, we might miss less severely impaired eL2 children (Tuller et al. 2018). To obtain a clearer picture of the usability of the LITMUS-QU-NWR, the findings should be replicated with bigger samples. Bigger samples would also enable us to compute the effects of additional factors, such as age, exposure, SES, and L1, in combination with each other.

Taken together, the results add further evidence that the short version of the LITMUS-QU-NWR (German) provides a suitable instrument that helps to disentangle TD and DLD

in eL2 learners and monolingual children aged 8 to 10. Despite these promising results, the LITMUS-QU-NWR should always be accompanied by other measures (for example sentence repetition, see Abed Ibrahim and Fekete 2019; Tuller et al. 2018; Armon-Lotem and Meir 2016) in order to detect DLD with high confidence.

**Funding:** This research was carried out in Project MILA which is part of the Research Center IDeA funded by the LOEWE program for excellency from the State of Hesse.

**Institutional Review Board Statement:** The study was conducted according to the guidelines of the Declaration of Helsinki and approved by the Ethics Committee of the German Psychological Association (DGPs) on 6 March 2009.

**Informed Consent Statement:** Informed consent was obtained from all subjects involved in the study.

**Data Availability Statement:** The present study is part of a larger research project. For data supporting the reported results, please consult the authors.

**Acknowledgments:** The authors thank Magdalena Wojtecka and Rabea Lemmer for their help managing the research project, Valentina Cristante for statistical support and Charlotte Lad for proofreading the manuscript. We are grateful to the children, their parents, and their kindergartens for participating in this study. We/I would like to thank the research assistants of project MILA for the data collection and analysis. We are very grateful to the editors and the reviewers and to Jan Schallenberger and Anna-Lena Scherger for their helpful comments on previous versions of the paper.

**Conflicts of Interest:** The author declares no conflict of interest. The funders had no role in the design of the study; in the collection, analysis, or interpretation of data; in the writing of the manuscript; or in the decision to publish the results.

### Appendix A. List of Items

*language-independent part:* pilu, kapi, lafi, faku, pli, kip, paklu, fluka, kafip, pukif, kifapu, kupafli, klipafu, flipuka, piklafu, kuflapi, kapufip, pifakup, flukif, klifak

*language-dependent part:* sapi, ʃaku, kiʃ, kas, sfupli, ʃpluki, sklifu, ʃlaklu, sklaplu, skifup, skifapu, ʃpafika, sfikupla, sklipafu, skaflipu, skapifuk, ʃpifakup, pukifs, fikapuks, kufiski

### Notes

[1] As pointed out by the authors, the better performances of Bi-DLD over Mo-DLD children have to do with the recruitment and the severity of the impairment.

[2] Although Plante and Vance (1995) do not mention this, their argument implies that a specificity below 70% can be considered as poor.

[3] Armon-Lotem and Meir (2016) find better diagnostic values for bilingual over monolingual children aged 5 to 6. The authors argue for a bilingual advantage in phonetic learning and/or higher experience with unfamiliar sound sequences (p. 729).

[4] It is an open question how spoken and written language deficits are related to each other. Following Bishop and Clarkson (2003), we assume that spoken and written language difficulties share the same underlying deficit.

[5] According to the authors, no differences exist between monolinguals and bilinguals in the norming sample.

[6] The effect of language group approaches significance.

[7] For children younger than age 5, there will be floor effects because even TD children will not have acquired the clusters of the language-dependent part.

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
