# Peer review of "The Use of the LITMUS Quasi-Universal Nonword Repetition Task to Identify DLD in Monolingual and Early Second Language Learners Aged 8 to 10"

_languages, doi:10.3390/languages7030218_

Round 1
Reviewer 1 Report
Summary
The manuscript entitled The Use Auf the LITMUS Quasi-Universal Nonword Repetition Task To Identify DLD in Monolingual and Early Second 3 Language Learners Aged 8 to 10 reports a study exploring the diagnostic effectiveness of a NWR task in identifying DLD in bilingual and monolingual children aged 8-10 in Germany. The NWR performance of 72 participants aged 8-10 and divided into 4 groups (monolingual with TD, monolingual with DLD, early successive bilingual with TD, early successive bilingual with DLD) was examined. Language status (TD vs. DLD) was determined based on performance on one morphosyntactic task in German (i.e., the TROG-D for monolinguals and the LiSe-DaZ task for the bilinguals). Logistic regression analysis was used to determine any group effects on the task and ROC-analyses were used to evaluate sensitivity and specificity. Particular attention was paid to the difference between performance on language dependent and language independent task items. While diagnostic effectiveness was somewhat mixed, the authors concluded that this particular NWR task can be used in combination with other measures to identify DLD in bilinguals with age of onset between 2 and 3 and aged 8-10 at time of testing.
Overall recommendation
This study fills a gap in the (Litmus) NWR literature in terms of addressing the factors of age (at time of testing) and age of onset and thus has the potential to make a significant contribution to the literature. However, as I detail below, there are some methodological issues and organizational/structural/clarity issues that need to be addressed.
Review
General Concept comments
- Age of onset and language status (TD/DLD)
One major issue concerns the characteristics of the two bilingual groups (TD vs DLD). As I understand it (but there should more details in the ms), the bilingual groups were recruited similarly (presumably via word-of-mouth, social media, local schools, etc., though this should be stated explicitly) and were attributed DLD/TD status based on their performance on the LiSe-DaZ. The authors then report that “the eL2DLD had a significantly later the [sic] age of onset to German than the eL2TD children (t(24) = -15.03, p < .001) and were less exposed German (t(24) = 9.59, p <.001) (lines 396-397).” Couldn’t it be, then, that the eL2DLD simply performed weaker than the eL2TD on the LiSe-DaZ because they had less exposure to German? While I understand that there are no clear solutions to the various issues that the recruitment and clinical status of these bilingual children raise, and that this is a problem for everyone in the field, it seems to me that this exposure problem should be addressed more explicitly in the ms. For example, perhaps the eL2DLD - eL2TD difference in performance on the LiSe-DaZ is so stark that differences in LoE likely do not provide a sufficient explanation?
Furthermore, since age at time of testing, age of onset and length of exposure are generally calculated using the same formula, it is a bit surprising that there is no significant difference in age at time of testing between the eL2DLD and eL2TD.
- The LD – LI distinction
The original rationale behind the LD-LI distinction was compelling. However, the results reported in this study, in addition to the results that the authors cite, suggest to me that this distinction is currently a distraction (or at least, confusing). I think the following aspects lead to this confusion: (1) The phonological properties are considered as LD vs L1 can vary across different versions of NWR tasks (as the authors note), (2) It is not clear which category leads to better identification of DLD (there is confusion in the lit review) and it is unclear how to interpret the LD vs. LI results in the current ms (as evidenced by contradictory statements, see (1-2) below), and (3) There is generally little difference in performance (in either monolinguals or bilinguals) between LI and LD items.
I think the results of the current ms make it clear that a more qualitative item analysis is necessary in order to improve the diagnostic effectiveness of Litmus-NWR. What I would like to see discussed (and I think the authors are in position to do so in the discussion of the current ms, provided that there is enough space) is which particular phonological properties (beyond the LD-LI distinction) seem particularly difficult for children with DLD. Is it nonwords with more than one cluster (CCV.CCV)? If I understand correctly, both LD and LI items can include this shape. Is it possible that there are items in both lists that are particularly sensitive to DLD and that examining the lists separately lowers diagnostic accuracy?
A related issue is that is it unclear whether the LI and LD items are similar in length or number of clusters per word. For example, how many LI items (vs LD items) have more than one cluster? How many LD items (vs LI items) have 3 syllables? The full list of NWR items should be provided to help the reader better understand the nature of the stimuli in each condition.
Related to the LD-LI distinction is the issue of L1 influence. While any L1 effects are unclear, there should be some clarification in terms of how many L1s represented in a particular participant pool have overlapping phonological properties with the L2. For example, how many of the L1s in the current study allow sCCVCV; 'CCVCVCs?
Some apparently contradictory statements about interpretation of the results of the current study (outside of the ones that the authors acknowledge in their lit review):
(1) Line 626: Either test part alone is unapplicable to identify DLD in eL2 learners (LI part: LR+ 2.1, LR- 0.5; LD part: 2.5, LR- 0.4).
(2) Line 657: Despite these methodical decisions, the present study demonstrated that the LD part of the LITMUS-QU-NWR alone identified monolingual children with DLD with high confidence and that the LD part was still suggestive in the eL2 group. These findings let conclude that the tool is usable to identify DLD in 8- to 10-year old children.
- Structure of the argumentation
The authors identify interesting gaps in the NWR literature involving age and age of onset. However, the argumentation is hard to follow. Ideally, the background should be structured so as to highlight these gaps and what is known (or not) about the impact of these factors. In terms of age at time of testing, what would be the prediction for the performance of older bilingual children compared to younger ones on NWR? What ages did Schwob et al. (2021) and Somberg (2020) include? In terms of age of acquisition, is there any evidence from the existing literature that this should have some effect on NWR? If so, what?
As it is currently written, the ms spends too much time discussing the history of the LITMUS-NWR (e.g., the content in paragraphs 75-96), which does not seem particularly relevant. Is there any real need to bring up the pre-version/long version? I would suggest reorganizing the background in order to focus on age, age of onset, and the LD-LI distinction (or whatever phonological property is of interest).
Related to issues with argumentation, the authors mention the following predictions/expectations in the discussion section that are not clearly laid out in the background:
- “Consistent with our expectations, the effect of Group was significant in the LI part for the comparison MoDLD-MoTD” (line 475)
- “findings of this study are in line with the prediction of the LITMUS-QU-NWR stating that differences between monolingual children and eL2 learners will rather appear in the LD than in the LI part.” (lines 615-616)
Finally, a reorganization of the background may allow for better clarity that would help the reader to better understand statements such as “Given the methodical differences across the studies, the language status and the superior discrimination of the LD part seem to be stable effects for children aged 5 to 9 years.” (lines 158 – 159) and “In sum, the studies so far provide promising results regarding the diagnostic usability of the LITMUS-QU-NWR for children aged 5 to 9.” (lines 230-231). As it is currently written, the evidence provided to justify such statements appears slim.
Specific (line-by-line) comments
There are a number of grammatical/stylistic issues throughout the ms. I note a number of them here:
Ln 39: also discriminate in big samples and long item lists. (reformulate)
Line 79: In the COST Action, two major strategies of constructing the nonwords have been pursued (Chiat 2015). -> were pursued
Line 125: age 5 and 9 are best studied for the (best? reformulate)
Line 214: because it considered the short version of the test and in part eL2 learners (in part?)
Line 248: In consequence, the experts tend to postpone the decision and a referral to a speech-language intervention ->as a consequence ; experts? = clinicians??
Please carefully define ‘different types of bilinguals’ (=in terms of age of onset?) and ‘language status’ (TD vs. DLD?) and use consistently
Line 369: visited German primary schools -> attended?
Line 421: considered in this study ->included in this study
Line 426: notebook? ->laptop?
Line 436: No transcriber had contact to the participants. (reformulate)
Line 581: differences to -> differences with
Reference list: please double check uniformity with respect to capitalization and punctuation. There are also a number of entries that appear incomplete (e.g. lines 763-765)
Author Response
Response to Reviewer 1
Dear Reviewer,
thank you for the careful reading and the plenty and helpful comments on our manuscript. We have revised the paper substantially. We are convinced that it improved a lot on the basis of your suggestions. Before we address the individual points of the review, we would like to start by summarizing the main changes:
- The reference standard is changed; classification is now based on referral and a language test
- The statistical analysis (mixed model) is changed according to the suggestions of the reviewers.
- The group comparisons are now based on non-parametrical tests because the data is not normally distributed.
- The manuscript has been proofread by a native speaker of English
Below we take the suggestion point by point.
Reviewer 1
Summary
The manuscript entitled The Use Auf the LITMUS Quasi-Universal Nonword Repetition Task To Identify DLD in Monolingual and Early Second 3 Language Learners Aged 8 to 10 reports a study exploring the diagnostic effectiveness of a NWR task in identifying DLD in bilingual and monolingual children aged 8-10 in Germany. The NWR performance of 72 participants aged 8-10 and divided into 4 groups (monolingual with TD, monolingual with DLD, early successive bilingual with TD, early successive bilingual with DLD) was examined. Language status (TD vs. DLD) was determined based on performance on one morphosyntactic task in German (i.e., the TROG-D for monolinguals and the LiSe-DaZ task for the bilinguals). Logistic regression analysis was used to determine any group effects on the task and ROC-analyses were used to evaluate sensitivity and specificity. Particular attention was paid to the difference between performance on language dependent and language independent task items. While diagnostic effectiveness was somewhat mixed, the authors concluded that this particular NWR task can be used in combination with other measures to identify DLD in bilinguals with age of onset between 2 and 3 and aged 8-10 at time of testing.
Overall recommendation
This study fills a gap in the (Litmus) NWR literature in terms of addressing the factors of age (at time of testing) and age of onset and thus has the potential to make a significant contribution to the literature. However, as I detail below, there are some methodological issues and organizational/structural/clarity issues that need to be addressed.
Review
General Concept comments
- Age of onset and language status (TD/DLD)
One major issue concerns the characteristics of the two bilingual groups (TD vs DLD). As I understand it (but there should more details in the ms), the bilingual groups were recruited similarly (presumably via word-of-mouth, social media, local schools, etc., though this should be stated explicitly)
>> Thank you for the comment. We agree with you that this is important and added more information about the recruitment.
“The monolinguals and eL2 learners were recruited between 2011 and 2012 in local schools, day-care centres and via SLTs. The recruitment took place as follows: In a first step, we sent an information letter to the institutions and asked for their participation. In case of agreement, we asked the teachers and SLTs to distribute further information letters and consent forms to the parents of monolingual and eL2 learners aged 8-10. If the parents agreed to participate, we contacted them by telephone and conducted an interview based on a questionnaire.”
We hope that these changes make clear that there was the same way of recruitment for the two groups.
and were attributed DLD/TD status based on their performance on the LiSe-DaZ.
>> No, different testes were used for monolinguals and eL2 learners. The monolinguals were diagnosed via the TROG-D (standardized for MON of this age group), and the eL2 learners with LiSe-DaZ. We reformulated this paragraph and motivate the choice of the tests:
“The classification as TD vs. DLD (= reference standard) is based on a) the referral to a speech-language intervention and b) on the performance in a language test (Bossuyt et al. 2015; Dollaghan and Horner 2011). Children are classified as TD if they were never referred to a speech-language therapy according to parental information and if they perform age-appropriately in the respective language test. Likewise, children are classified as DLD if they were referred to speech-language therapy and if they perform T < 40 in the language test. In the monolingual group, there is a perfect match of referral and result in the TROG-D, i.e. no over- or underdiagnosis. In the eL2 group, out of the original sample of n = 36 eL2 learners, referral and test results match in 33/36 (91.7%) children. The three remaining children all are underdiagnosed (i.e. not referred despite a poor test result). These three children are excluded from the analysis, resulting in n = 33 eL2 learners.
As stated above, different language tests are used for the monolingual and the eL2 groups. In the monolingual group, we use the TROG-D (Fox-Boyer 2016), a standardized language test assessing sentence comprehension. The test provides norms for monolinguals aged 3;0 to 10;11 years and only has to be administered to monolingual children. Monolingual children are classified as DLD if they score T < 40 in the TROG-D and if they were referred to speech-language therapy.
For the eL2 group aged 8 to 10, no comparable norm-referenced language test is available. We choose LiSe-DaZ, a standardized language test which has been constructed with a particular focus on eL2 learners, for two reasons: First, due to the focus on eL2 learners, LiSe-DaZ is culturally less biased than tests developed for monolingual children (e.g. the TROG-D). Second, LiSe-DaZ provides norms for eL2 learners between 3;6 and 7;11 years. We adopt these norms for our older participants and classify eL2 learners as DLD if they score T < 40 in two or more subtests of LiSe-DaZ, and if they were referred to speech-language therapy. Given the range of L1s, no testing in the L1 is possible, but the onset of the single word and the multiword stage are considered as additional confirmation (see below for more information).”
The authors then report that “the eL2DLD had a significantly later the [sic] age of onset to German than the eL2TD children (t(24) = -15.03, p < .001) and were less exposed German (t(24) = 9.59, p <.001) (lines 396-397).” Couldn’t it be, then, that the eL2DLD simply performed weaker than the eL2TD on the LiSe-DaZ because they had less exposure to German? While I understand that there are no clear solutions to the various issues that the recruitment and clinical status of these bilingual children raise, and that this is a problem for everyone in the field, it seems to me that this exposure problem should be addressed more explicitly in the ms. For example, perhaps the eL2DLD - eL2TD difference in performance on the LiSe-DaZ is so stark that differences in LoE likely do not provide a sufficient explanation?
>> We considered only exposure because AoO and exposure are normally strongly correlated in eL2 children. Therefore, the model didn’t converge. For the model, we then choose exposure because we supposed that exposure matters more than AoO.
Following the suggestions of another reviewer, we checked the distribution of the data and now run non-parametrical tests. In connection with re re-classification due to the new reference standard (referral & language test), only SES turned out to be significant.
Nevertheless, as pointed out for example by Tuller et al (2018), the absence of an effect of AoO or exposure does not mean that AoO or exposure play no role at all. In fact, descriptively, the AoO is three months later in the eL2DLD group (but exposure is almost identical; 73.3 vs. 73.1 months). Given the relatively long experience to German of more than 6 years, we think that the differences in AoO will not have an effect. Therefore, the difficulties in the DLD group are due to language problems, and not (or very little) due other factors.
Furthermore, since age at time of testing, age of onset and length of exposure are generally calculated using the same formula, it is a bit surprising that there is no significant difference in age at time of testing between the eL2DLD and eL2TD.
>> That was the outcome of the statistics. However, it has changed in the new version (see above).
- The LD – LI distinction
The original rationale behind the LD-LI distinction was compelling. However, the results reported in this study, in addition to the results that the authors cite, suggest to me that this distinction is currently a distraction (or at least, confusing). I think the following aspects lead to this confusion: (1) The phonological properties are considered as LD vs L1 can vary across different versions of NWR tasks (as the authors note), (2) It is not clear which category leads to better identification of DLD (there is confusion in the lit review) and it is unclear how to interpret the LD vs. LI results in the current ms (as evidenced by contradictory statements, see (1-2) below), and (3) There is generally little difference in performance (in either monolinguals or bilinguals) between LI and LD items.
I think the results of the current ms make it clear that a more qualitative item analysis is necessary in order to improve the diagnostic effectiveness of Litmus-NWR.
>> Thank you for the comment. We agree with you that it is important to know more about – and to have empirical findings on this question - which qualitative properties account for the good diagnostic usability of the test. We will address this point in future work which is already in planning. However, this paper is aimed to replicate and expand quantitative findings: group differences and diagnostic accuracy.
What I would like to see discussed (and I think the authors are in position to do so in the discussion of the current ms, provided that there is enough space) is which particular phonological properties (beyond the LD-LI distinction) seem particularly difficult for children with DLD. Is it nonwords with more than one cluster (CCV.CCV)? If I understand correctly, both LD and LI items can include this shape. Is it possible that there are items in both lists that are particularly sensitive to DLD and that examining the lists separately lowers diagnostic accuracy?
>> Thank you for this comment. This is an interesting point, see our response above. For space reasons, we cannot go into more detail with regard to the question which items discriminate in which group. We refer to a recent analysis by Schallenberger (2021) who found that different items showed the optimal discriminatory values, depending on age and language group.
A related issue is that is it unclear whether the LI and LD items are similar in length or number of clusters per word.
>> Thank you for the comment. The focus of the test is to allow group comparisons, namely TD vs. DLD. The both parts were never designed to be compared directly because they cannot be balanced with regard to complexity: As we point out, the LD part is inherently more complex than the LI part because 3-memebr clusters are only possible in this part. Therefore, we didn’t attempt to balance length or number of clusters between the two test parts.
For example, how many LI items (vs LD items) have more than one cluster? How many LD items (vs LI items) have 3 syllables? The full list of NWR items should be provided to help the reader better understand the nature of the stimuli in each condition.
>> See the appendix for the list of items. More details about the construction of the items are given in previous work that we mention (e.g. dos Santos & Ferré 2016, Ferré and dos Santos 2015).
Related to the LD-LI distinction is the issue of L1 influence. While any L1 effects are unclear, there should be some clarification in terms of how many L1s represented in a particular participant pool have overlapping phonological properties with the L2. For example, how many of the L1s in the current study allow sCCVCV; 'CCVCVCs?
>> Thank you for this comment. We agree with you that it is necessary to know more about potential effects of the L1. However, given the variety of L1s in our sample, it is virtually impossible to consider the structure of all these languages.
We take up the question of potential L1 effects at several places in the paper in the Intro and Discussion. On the one hand, we refer to previous work that systematically compared children with different L1s (e.g. Chilla et al. 2021), and who found no effect. On the other, we add a few sentences on the aquisition of phonology in early bilinguals (in phonology ‘early’ normally means before age 6). This should make clear that L1 effects are presumably are not fully absent at age 8 to 10, but cannot account for the TD-DLD differences the paper focuses on.
“A few studies on the LITMUS-QU-NWR take additional factors like L1, age, age of onset, exposure, and SES into account (Chilla et al. 2021; Almeida et al. 2017; Tuller et al. 2018). The L1 does not significantly affect the performance in the LITMUS-QU-NWR (Chilla et al. 2021 for a comparison of children with L1 Arabic, Turkish, and Portuguese). This finding is in line with the general observation that, in early learners, L1 effects are transient (Fennell and Tsui 2020), are characteristic of the early stages of phonological development (Kehoe 2015), and that differences to monolinguals disappear after “a couple of years” (Holm and Dodd 2006: 307).“
Some apparently contradictory statements about interpretation of the results of the current study (outside of the ones that the authors acknowledge in their lit review):
(1) Line 626: Either test part alone is unapplicable to identify DLD in eL2 learners (LI part: LR+ 2.1, LR- 0.5; LD part: 2.5, LR- 0.4).
(2) Line 657: Despite these methodical decisions, the present study demonstrated that the LD part of the LITMUS-QU-NWR alone identified monolingual children with DLD with high confidence and that the LD part was still suggestive in the eL2 group. These findings let conclude that the tool is usable to identify DLD in 8- to 10-year old children.
>> We hope that these contradictions will be resolved under the new results.
- Structure of the argumentation
The authors identify interesting gaps in the NWR literature involving age and age of onset. However, the argumentation is hard to follow. Ideally, the background should be structured so as to highlight these gaps and what is known (or not) about the impact of these factors. In terms of age at time of testing, what would be the prediction for the performance of older bilingual children compared to younger ones on NWR?
>> Thank you for this comment. We have re-formulated the predictions (see section rationale and research questions).
What ages did Schwob et al. (2021) and Somberg (2020) include?
>> Thank you for pointing this out. The age ranges in Somberg (2020) and of both meta-studies are added.
In terms of age of acquisition, is there any evidence from the existing literature that this should have some effect on NWR? If so, what?
>> Yes, there are contradictory findings on the role of exposure. We have added a paragraph to make this (more) clear.
“Within the bilingual groups, factors of language exposure or language use were not significantly related to the performance in the LITMUS-QU-NWR (Tuller et al. 2018; Almeida et al. 2017; Chilla et al. 2021), but factors such as positive early development, and to a lower extent, age could modulate differences within the bilingual group (see Tuller et al. 2018; Almeida et al. 2017 for a discussion). Comparing 2L1, eL2 and lL2 learners by their age of onset, Somberg (2020) finds no group differences. SES plays a minor role for the LITMUS-QU-NWR (Tuller et al. 2018), however, as pointed out by Tuller et al. (2018), small effects might have been undetected due to the limited sample size and limited range in age and exposure. Results from other NWR’s suggest that chronological age and exposure can influence the performance (e.g., Duncan and Paradis 2016; Thordardottir and Brandeker 2013), and that SES is neglectable in connection with nonword repetition (Chiat and Roy 2007; Boerma et al. 2015; Engel de Abreu, F. H. dos Santos, and Gathercole 2008; Chiat and Polišenská 2016; but see Meir and Armon-Lotem 2017).“
Also, a later onset is correlated with less exposure to the L2. Therefore, eL2-TD learners can be outperformed by Mo-TD, at leats in the LD part. As we point out at several places in the article, most studies so far summarized simultaneous and successive bilingual learners. Effects of Age of onset or exposure might have been covered by the composition and/or size of the groups. Our study aims to take a closer look on the performance in eL2 learners.
As it is currently written, the ms spends too much time discussing the history of the LITMUS-NWR (e.g., the content in paragraphs 75-96), which does not seem particularly relevant. Is there any real need to bring up the pre-version/long version? I would suggest reorganizing the background in order to focus on age, age of onset, and the LD-LI distinction (or whatever phonological property is of interest).
>> We have added information about effects of factors like age, age of onset, L1, etc. However, we feel that it is important to highlight the development of the test (and will not remove it) because there is some confusion about the two versions within the LITMUS community and among researchers who plan to use the material.
Related to issues with argumentation, the authors mention the following predictions/expectations in the discussion section that are not clearly laid out in the background:
- “Consistent with our expectations, the effect of Group was significant in the LI part for the comparison MoDLD-MoTD” (line 475)
- “findings of this study are in line with the prediction of the LITMUS-QU-NWR stating that differences between monolingual children and eL2 learners will rather appear in the LD than in the LI part.” (lines 615-616)
>> Thank you for this helpful comment. We take it up and hope that the predictions are more clear now.
Finally, a reorganization of the background may allow for better clarity that would help the reader to better understand statements such as “Given the methodical differences across the studies, the language status and the superior discrimination of the LD part seem to be stable effects for children aged 5 to 9 years.” (lines 158 – 159)
and “In sum, the studies so far provide promising results regarding the diagnostic usability of the LITMUS-QU-NWR for children aged 5 to 9.” (lines 230-231). As it is currently written, the evidence provided to justify such statements appears slim.
>> The respective sections are reformulated. We hope that it is easier to understand.
Specific (line-by-line) comments
There are a number of grammatical/stylistic issues throughout the ms. I note a number of them here:
Ln 39: also discriminate in big samples and long item lists. (reformulate)
>> done: Mono- and bisyllabic items discriminate if big samples are examined and/or long item lists are used (‘short item effect’).
Line 79: In the COST Action, two major strategies of constructing the nonwords have been pursued (Chiat 2015). -> were pursued
>> done.
Line 125: age 5 and 9 are best studied for the (best? reformulate)
>> We are no native speaker of English, but the native English proof-reader didn’t mark this as ungrammatical. Nevertheless, we hope that the reformulation sounds better: “To date, children between age 5 and 9 are most studied for the LITMUS-QU-NWR… ”
Line 214: because it considered the short version of the test and in part eL2 learners (in part?)
>> Thank you. This section is reformulated.
Line 248: In consequence, the experts tend to postpone the decision and a referral to a speech-language intervention ->as a consequence ; experts? = clinicians??
>> We use the passive: “As a consequence, the decision and a referral to speech-language therapy is postponed (Voet Cornelli 2020).”
In Germany, only doctors are allowed to refer to a therapy, not SLTs, but without a further explanation, we feel that ‘the doctors’ come ad hoc.
Please carefully define ‘different types of bilinguals’ (=in terms of age of onset?) and ‘language status’ (TD vs. DLD?) and use consistently
>> done
Line 369: visited German primary schools -> attended?
>> done
Line 421: considered in this study ->included in this study
>> done
Line 426: notebook? ->laptop?
>> done
Line 436: No transcriber had contact to the participants. (reformulate)
>> done: … student assistant who was not the investigator.
Line 581: differences to -> differences with
>> done
Reference list: please double check uniformity with respect to capitalization and punctuation. There are also a number of entries that appear incomplete (e.g. lines 763-765)
>> done

Reviewer 2 Report
Manuscript title: The Use Auf the LITMUS Quasi-Universal Nonword Repetition Task To Identify DLD in Monolingual and Early Second 3 Language Learners Aged 8 to 10
Brief summary: The study evaluates effects of clinical status (DLD vs. TLD) and language status (mono vs. bi) on the performance of LITMUS QU NWR task comprised of language-dependent and language-independent subtasks in German-speaking children aged 8-10 (n=76: 36 monolinguals, 36 eL2 learners). The results for the language-independent and the language-dependent parts showed significant differences between the clinical groups (TLD vs. DLD), yet no differences between monolingual and bilinguals. The evaluation of the diagnostic accuracy as determined by ROC analysis revealed that cut-off points are different for monolingual and bilinguals for the language-dependent part yet overlap for the language-independent. The likelihood ratios turned out to be good to suggestive for monolinguals, yet neither part of the test adequately discriminated between the two clinical groups in bilinguals.
Overall evaluation: I have greatly enjoyed reading the referred study and therefore I warmly recommend it to be published. The study has important clinical implications, and it fits the special issue theme.
Having said that I would like the authors to consider the following points when preparing a revised version of the manuscript. Points 1 and 2 are more conceptual / methodological. While points presented in a page-by-page manner are minor and technical.
- Terminology: I found it a bit confusing that the authors chose the term “language status” to refer to DLD-vs.-TLD differences. I would recommend to choose “clinical status” when referring to DLD-vs.-TLD differences, and “language status” when referring to mono-vs.-eL2 differences.
- 8 “No child had to be excluded due to a low nonverbal IQ (i.e. below 80).” => it should be added in a footnote that the current DLD diagnosis does not make reference to the non-verbal IQ, in contrast to the previous criteria for the diagnosis of SLI. See
Bishop, D. V. (2017). Why is it so hard to reach agreement on terminology? The case of developmental language disorder (DLD). International journal of language & communication disorders, 52(6), 671-680.
Bishop, D. V., Snowling, M. J., Thompson, P. A., Greenhalgh, T., Catalise‐2 Consortium, Adams, C., ... & house, A. (2017). Phase 2 of CATALISE: A multinational and multidisciplinary Delphi consensus study of problems with language development: Terminology. Journal of Child Psychology and Psychiatry, 58(10), 1068-1080.
Abstract, methodology and p. 12 “The present study explored if the short version of the German LITMUS-QU-NWR can be used to disentangle DLD 8 to 10-year old children classified by a language test.”
Diagnosis of DLD: I think that a clarification on the diagnosis of DLD in eL2 is needed. I did not find any information in the text with respect to children`s skills in L1. Has any data collected regarding milestones in L1 (parental concern regarding L1 and L2 / parental rating of L1 and L20, I understand that it is virtually impossible to test all L1`s of the children via standardized assessment tools. Yet, I believe that it is vital to refer to L1 development. DLD in bilinguals is diagnosed based on poor performance in BOTH languages. We must rule out the concern that some of these children are NOT DLD, yet L1 dominant and show low performance in L2-German.
- Statistical Analysis.
Background information. When presenting background information (see p.8), I wonder if the lengthy description of the groups (e.g., gender / age / length of schooling) should be presented in a table for the convenience of a reader with means (SDs) and ranges in order to make the information more accessible. When analyzing background information for the 4 groups, I would recommend to run ANOVAS and post-hoc pair-wise comparisons with corrected alpha-levels. I would strongly avoid running simple t-tests as currently presented on p. 8-9.
NWR performance. I absolutely agree that the binomial mixed effects regression is the most optimal analysis. Yet, I wonder if the data should be analyzed using “language status: DLD-vs.-TLD and “clinical status: mono-vs.-bi” as well as their interaction as fixed variables. If interactions improve the fit of the models, then emeans functions are to be to be applied in order to unpack the source of the interaction. For transparency, the R formulas should be added.
Regarding random effects, it is not clear whether they were entered as two random intercepts. Did the authors include random slopes for participants and items? It might be the case that the model did not converge, yet it should be stated that the model did not converge, and/ or did not significantly improve the fit of the model.
Furthermore, I would recommend providing the final model for the two analyses in the results subsection, rather than in the Appendix (see Appendix A).
Minor page-by-page comments
Title: Manuscript title: The Use Auf the LITMUS Quasi-Universal Nonword Repetition Task To Identify DLD in Monolingual and Early Second 3 Language Learners Aged 8 to 10
- The use “of”….
The use of abbreviations: I would recommend decreasing the number of abbreviations, even for me who is very familiar with the field, it was challenging to read the manuscript due to a large number of abbreviations. First, I would not use LI and LD as abbreviations of the NWR subtests, as these abbreviations might stand for Language Impairment / Language Disorder, which are discussed at large in the manuscript. I would recommend using “language-independent” and “language-dependent” equivalent, which should not change the word count. With respect to groups, I also found it a bit challenging to follow their abbreviations due to large number of consonants (e.g., eL2DLD), maybe the use of hyphens might help. Consider: eL2-DLD / MO-DLD/ MO-TD /eL2-TD.
Abstract: the authors refer to complexity in the abstract several times, yet, the study does not deal per se with the complexity. It does deal with group differences and the diagnostic accuracy. Therefore, I would recommend revising the abstract to better reflect the RQs of the study.
p.2 “Trisyllabic and longer items seem to discriminate better than shorter items but items up to three syllables also discriminate in big samples and long item lists.” => re-word / not clear
p.2 “The meta-study 47 of Ortiz (2021) is based on 13 studies including bilinguals and focused on the diagnostic accuracy of NWR tests.” => re-word / not clear
p.2 With regard to the present study, the most important findings of the two meta-analyses are a) that there were consistent effects of languages status (TD vs. SLI/DLD), b) that there were larger effect sizes and better discrimination for language-independent compared to language-specific tests, and c) that many of the NWR tests, in particular the language-independent tests, showed good to very good diagnostic values. => what about mono-bi differences, the study does discuss this difference, so previous research should be also summarized here.
p.4 “Most studies on the LITMUS-QU-NWR examined the whole word accuracy” => Should “whole word accuracy” be replaced with “whole nonword accuracy” / “whole item accuracy”?
p.5 “This has to do with different location of recruitment in monolinguals 208 and bilinguals (dos Santos and Ferré 2016), different severities of the disease… ” => replace “disease” with “disorder”
p.6 “Among other factors (e.g., the lack of appropriate tools, see Almeida et al. 2017; Tuller et al. 2018), this has to do with the fact 245 that at the compulsory language screenings, many bilinguals, and in particular successive learners, have too little exposure in the L2 to allow a decision on whether the child is language-impaired not. “ => or is missing before “not”.
P8. We decided to use LiSe-DaZ (Schulz and Tracy 2011) for the eL2 group because LiSe-DaZ is culturally fair. => culturally unbiased?
p.13 “This suggests that discrimination is not perfect.” => reword it. “This suggests that the diagnostic accuracy is not ideal / not acceptable”.
p.14 “Moreover, a better match between diagnosis (via tests) and identification (via the LIT-641 MUS-QU-NWR) is likely if the diagnosis includes a phonological test because both 642 measures tap into phonological abilities” => not clear
p.15 “In our study, we found no effects of SES (measured as mother’s length of 670 schooling), in line with research on other NWRs (Schwob et al. 2021; Tuller et al. 2018).” See these studies which were designed assess effects of SES and bilingualism:
Meir, N., & Armon-Lotem, S. (2017). Independent and combined effects of socioeconomic status (SES) and bilingualism on children’s vocabulary and verbal short-term memory. Frontiers in Psychology, 8, 1442.
Chiat, S., & Polišenská, K. (2016). A framework for crosslinguistic nonword repetition tests: Effects of bilingualism and socioeconomic status on children's performance. Journal of Speech, Language, and Hearing Research, 59(5), 1179-1189.
Author Response
Response to Reviewer 2
Dear Reviewer
We would like to thank you very much for the careful reading and the plenty and very helpful comments to our manuscript. We have revised the paper substantially, and we are convinced that it improved a lot on the basis of your suggestions. Before we address the individual points of the review, we would like to start by summarizing the main changes:
- The reference standard is changed; classification is now based on referral and a language test
- The statistical analysis (mixed model) is changed according to the suggestions of the reviewers.
- The group comparisons are now based on non-parametrical tests because the data is not normally distributed.
- The manuscript has been proofread by a native speaker of English
Below we take the suggestion point by point.
Manuscript title: The Use Auf the LITMUS Quasi-Universal Nonword Repetition Task To Identify DLD in Monolingual and Early Second 3 Language Learners Aged 8 to 10
Brief summary: The study evaluates effects of clinical status (DLD vs. TLD) and language status (mono vs. bi) on the performance of LITMUS QU NWR task comprised of language-dependent and language-independent subtasks in German-speaking children aged 8-10 (n=76: 36 monolinguals, 36 eL2 learners). The results for the language-independent and the language-dependent parts showed significant differences between the clinical groups (TLD vs. DLD), yet no differences between monolingual and bilinguals. The evaluation of the diagnostic accuracy as determined by ROC analysis revealed that cut-off points are different for monolingual and bilinguals for the language-dependent part yet overlap for the language-independent. The likelihood ratios turned out to be good to suggestive for monolinguals, yet neither part of the test adequately discriminated between the two clinical groups in bilinguals.
Overall evaluation: I have greatly enjoyed reading the referred study and therefore I warmly recommend it to be published. The study has important clinical implications, and it fits the special issue theme.
Having said that I would like the authors to consider the following points when preparing a revised version of the manuscript. Points 1 and 2 are more conceptual / methodological. While points presented in a page-by-page manner are minor and technical.
- Terminology: I found it a bit confusing that the authors chose the term “language status” to refer to DLD-vs.-TLD differences. I would recommend to choose “clinical status” when referring to DLD-vs.-TLD differences, and “language status” when referring to mono-vs.-eL2 differences.
>> Thank you for this comment. We have changed the terms: language group refers to monolingual vs. eL2 and clinical status to TD vs. DLD.
- 8 “No child had to be excluded due to a low nonverbal IQ (i.e. below 80).” => it should be added in a footnote that the current DLD diagnosis does not make reference to the non-verbal IQ, in contrast to the previous criteria for the diagnosis of SLI. See
Bishop, D. V. (2017). Why is it so hard to reach agreement on terminology? The case of developmental language disorder (DLD). International journal of language & communication disorders, 52(6), 671-680.
Bishop, D. V., Snowling, M. J., Thompson, P. A., Greenhalgh, T., Catalise‐2 Consortium, Adams, C., ... & house, A. (2017). Phase 2 of CATALISE: A multinational and multidisciplinary Delphi consensus study of problems with language development: Terminology. Journal of Child Psychology and Psychiatry, 58(10), 1068-1080.
>> Thank you very much for the suggestion. We added these two publications and the respective information in the recruitment section : “Based on the definition of DLD (Bishop 2017; Bishop et al. 2017), the IQ provides no exclusionary criterion.”
Abstract, methodology and p. 12 “The present study explored if the short version of the German LITMUS-QU-NWR can be used to disentangle DLD 8 to 10-year old children classified by a language test.”
Diagnosis of DLD: I think that a clarification on the diagnosis of DLD in eL2 is needed. I did not find any information in the text with respect to children`s skills in L1. Has any data collected regarding milestones in L1 (parental concern regarding L1 and L2 / parental rating of L1 and L20, I understand that it is virtually impossible to test all L1`s of the children via standardized assessment tools. Yet, I believe that it is vital to refer to L1 development. DLD in bilinguals is diagnosed based on poor performance in BOTH languages. We must rule out the concern that some of these children are NOT DLD, yet L1 dominant and show low performance in L2-German.
>> Thank you very much for rising this important point. We have taken up this point in two ways.
- We now consider referral and performance in a language test as reference standard (see section Classification as TD or DLD):
“Children are classified as TD if they were never referred to a speech-language therapy according to parental information and if they perform age-appropriately in the respective language test. Likewise, children are classified as DLD if they were referred to speech-language therapy and if they perform T < 40 in the language test. In the monolingual group, there is a perfect match of referral and result in the TROG-D, i.e. no over- or underdiagnosis. In the eL2 group, out of the original sample of n = 36 eL2 learners, referral and test results match in 33/36 (91.7%) children. The three remaining children all are underdiagnosed (i.e. not referred despite a poor test result). These three children are excluded from the analysis, resulting in n = 33 eL2 learners. “
Note that the now reference standard resulted in a re-classification of the eL2 learners and exclusion of three children. Monolinguals were not affected because there was a perfect match between referral and test.
- We explicitly state that no direct assessment in the L1 took place.
- We now consider the early milestones as additional indicators of (non-)DLD. We tried to point out that these factors are associated with the abilities in the L1 in eL2 learners. Given that the age of onset is factually around age 3, the first words and word combinations should be produced in the L1, at least in eL2-TD children and by most of the eL2DLD children. In fact, some eL2DLD children produced the first words and first word combinations around age 3, but independently of the language (i.e. if first words and word combinations were produced in the L1 or L2), this is a clear indication of an impairment. We considered it more informative to present the qualitative and quantitative information for the two indicators separately instead of calculating an early risk index (e.g. Boerma & Blom 2017; Tuller et al. 2018) because no regression analysis was planned.
- Statistical Analysis.
Background information. When presenting background information (see p.8), I wonder if the lengthy description of the groups (e.g., gender / age / length of schooling) should be presented in a table for the convenience of a reader with means (SDs) and ranges in order to make the information more accessible. When analyzing background information for the 4 groups, I would recommend to run ANOVAS and post-hoc pair-wise comparisons with corrected alpha-levels. I would strongly avoid running simple t-tests as currently presented on p. 8-9.
NWR performance. I absolutely agree that the binomial mixed effects regression is the most optimal analysis. Yet, I wonder if the data should be analyzed using “language status: DLD-vs.-TLD and “clinical status: mono-vs.-bi” as well as their interaction as fixed variables. If interactions improve the fit of the models, then emeans functions are to be to be applied in order to unpack the source of the interaction. For transparency, the R formulas should be added.
>> Thank you for this helpful suggestion. We have changed the model accordingly.
Regarding random effects, it is not clear whether they were entered as two random intercepts. Did the authors include random slopes for participants and items? It might be the case that the model did not converge, yet it should be stated that the model did not converge, and/ or did not significantly improve the fit of the model.
>> We entered the random effects as random intercepts without including random slopes for participants and items. In the description of the analysis we have now added the formula of the model: accuracy ~ clinic status * language group + SES + (1 | subject) + (1 | item), family = binomial.
Furthermore, I would recommend providing the final model for the two analyses in the results subsection, rather than in the Appendix (see Appendix A).
>> The final model is now placed in the results section, see tables 5 and 6.
Minor page-by-page comments
Title: Manuscript title: The Use Auf the LITMUS Quasi-Universal Nonword Repetition Task To Identify DLD in Monolingual and Early Second 3 Language Learners Aged 8 to 10
- The use “of”….
>> Thank you. The preposition is added.
The use of abbreviations: I would recommend decreasing the number of abbreviations, even for me who is very familiar with the field, it was challenging to read the manuscript due to a large number of abbreviations. First, I would not use LI and LD as abbreviations of the NWR subtests, as these abbreviations might stand for Language Impairment / Language Disorder, which are discussed at large in the manuscript. I would recommend using “language-independent” and “language-dependent” equivalent, which should not change the word count. With respect to groups, I also found it a bit challenging to follow their abbreviations due to large number of consonants (e.g., eL2DLD), maybe the use of hyphens might help. Consider: eL2-DLD / MO-DLD/ MO-TD /eL2-TD.
>> Thank you. I fully agree with you and gladly take up your suggestions. The abbreviations LI and LD are used only in the tables and figures for space limitations.
Abstract: the authors refer to complexity in the abstract several times, yet, the study does not deal per se with the complexity. It does deal with group differences and the diagnostic accuracy. Therefore, I would recommend revising the abstract to better reflect the RQs of the study.
>> Done. Thank you!
p.2 “Trisyllabic and longer items seem to discriminate better than shorter items but items up to three syllables also discriminate in big samples and long item lists.” => re-word / not clear
>> The sentence has been reformulated: Mono- and bisyllabic items discriminate if big samples are examined and/or long item lists are used (‘short item effect’).
p.2 “The meta-study 47 of Ortiz (2021) is based on 13 studies including bilinguals and focused on the diagnostic accuracy of NWR tests.” => re-word / not clear
>> reformulated: ‘The meta-study of Ortiz (2021) is based on 13 studies (6 of them including only bilinguals, 7 including both monolinguals and bilinguals).’
p.2 With regard to the present study, the most important findings of the two meta-analyses are a) that there were consistent effects of languages status (TD vs. SLI/DLD), b) that there were larger effect sizes and better discrimination for language-independent compared to language-specific tests, and c) that many of the NWR tests, in particular the language-independent tests, showed good to very good diagnostic values. => what about mono-bi differences, the study does discuss this difference, so previous research should be also summarized here.
>>Thank you for this comment. We mention that there are monolingual-bilingual differences in NWR and refer literature that finds such differences:
“This stands in contrast to other studies using nonword repetition that report monolingual-bilingual differences in TD children (e.g. Gutiérrez-Clellen and Simon-Cereijido 2010).“
However, given the aims of the article, we would not elaborate more on these differences.
p.4 “Most studies on the LITMUS-QU-NWR examined the whole word accuracy” => Should “whole word accuracy” be replaced with “whole nonword accuracy” / “whole item accuracy”?
>> Thank you. I use ‘whole item accuracy’.
p.5 “This has to do with different location of recruitment in monolinguals 208 and bilinguals (dos Santos and Ferré 2016), different severities of the disease… ” => replace “disease” with “disorder”
>> Thank you. ‘desease’ was replaced by ‘disorder’ throughout the paper.
p.6 “Among other factors (e.g., the lack of appropriate tools, see Almeida et al. 2017; Tuller et al. 2018), this has to do with the fact 245 that at the compulsory language screenings, many bilinguals, and in particular successive learners, have too little exposure in the L2 to allow a decision on whether the child is language-impaired not. “ => or is missing before “not”.
>> done: ‘In particular successive-bilingual children have too little exposure to the L2 to allow a severe decision on whether the child is language-impaired or not.’
P8. We decided to use LiSe-DaZ (Schulz and Tracy 2011) for the eL2 group because LiSe-DaZ is culturally fair. => culturally unbiased?
>> done: ‘First, due to the focus on eL2 learners, LiSe-DaZ is culturally less biased than tests developed for monolingual children (e.g. the TROG-D).’
p.13 “This suggests that discrimination is not perfect.” => reword it. “This suggests that the diagnostic accuracy is not ideal / not acceptable”.
>> The sentence has been reformulated.
p.14 “Moreover, a better match between diagnosis (via tests) and identification (via the LIT-641 MUS-QU-NWR) is likely if the diagnosis includes a phonological test because both 642 measures tap into phonological abilities” => not clear
>> The whole section has been re-written.
p.15 “In our study, we found no effects of SES (measured as mother’s length of 670 schooling), in line with research on other NWRs (Schwob et al. 2021; Tuller et al. 2018).” See these studies which were designed assess effects of SES and bilingualism:
Meir, N., & Armon-Lotem, S. (2017). Independent and combined effects of socioeconomic status (SES) and bilingualism on children’s vocabulary and verbal short-term memory. Frontiers in Psychology, 8, 1442.
Chiat, S., & Polišenská, K. (2016). A framework for crosslinguistic nonword repetition tests: Effects of bilingualism and socioeconomic status on children's performance. Journal of Speech, Language, and Hearing Research, 59(5), 1179-1189.
>> Thank you for the references. We have added these studies in the respective paragraph:
“SES plays a minor role for the LITMUS-QU-NWR (Tuller et al. 2018), however, as pointed out by Tuller et al. (2018), small effects might have been undetected due to the limited sample size and limited range in age and exposure. Results from other NWR’s suggest that chronological age and exposure can influence the performance (e.g., Duncan and Paradis 2016; Thordardottir and Brandeker 2013), and that SES is neglectable in connection with nonword repetition (Chiat and Roy 2007; Boerma et al. 2015; Engel de Abreu, F. H. dos Santos, and Gathercole 2008; Chiat and Polišenská 2016; but see Meir and Armon-Lotem 2017).“

Reviewer 3 Report
The article focuses on the QU-NWRT (German). Studies on NWRTs are frequent and have already shown the potential of this kind of task to identify a DLD in a monolingual and bilingual population. However, The study is innovative in that it focuses on older children than in most NRWT research.
The structure of the article is adequate, easy for the reader. The introduced references are recent and provide a good overview of the subject and the studies on NWRT. The results are well presented. However, several general and specific comments have been incorporated to improve the article.
The weak points of the article are, in my opinion, the constitution and distribution of the TD and DLD groups, which then has an impact on the results of the study. Furthermore, the lack of inter-judge reliability for the analysis of the replicates is also a weak point. These elements can be further justified by the authors in the review.
Title:
- of vs. auf ?
Introduction :
- line 62: Which additional tools/measures can be used?
- line 125: Why are 5-9 year olds studied the most? Is this related to the age of diagnosis of DLD?
- line 155: Is it always counted according to the number of correct phonemes? There seem to be differences according to the tasks and preferences of the authors (e.g., Tuller et al., 2018)
- line 178: why not talk about the Youden indices (1950)? What are the differences with what you present in your paper?
- line 232: why is it interesting to look at data from 8-10 year olds? are there reasons? If so, they could be stated by the authors.
- line 240: Are children referred to speech and language therapy between the ages of 8-10 necessarily referred for oral language? Isn't the priority given to learning and written language at this age? you explain something about it in the following lines, but the question is already being asked here.
Method:
- line 328: why use SLI and DLD in a similar way? Focus on DLD and the CATALISE consensus of Bishop et al (2017).
- line 333: Is the questionnaire used based on the PabiQ (Tuller, 2015)? If so, it should be cited.
Or what does your questionnaire add? - line 348: why only a test of syntactic comprehension was used? So your DLD children must have had receptive difficulties? What about the different domains?
A diagnosis is not made by assessing only one domain. Furthermore, a diagnosis of DLD is made when there are persistent difficulties and functional impacts, what about them? Indeed, it does not seem certain that the children classified as DLD in the research actually have DLD. For these reasons, it seems complex to then perform sensitivity and specificity calculations on children whose categorisation is not clear at the beginning. - line 447: why not conduct an interrater reliability?
- Material: if possible, it would be good to add the list of items (LI and LD) in the supplementary material or appendix
Results:
- well presented, very clear
- but it might be a good idea to add the effect sizes in addition to the p. values
- line 501: I think it would be wise to explain how the cut-offs were calculated to make it easier for the reader to read and analyse.
Discussion:
- line 558: and also in relation to the small number of studies that looked at LI items included in the meta-analyses.
- line 657: The authors have well explained and justified the results obtained in terms of diagnostic accuracy.
However, in my opinion, this could have been partly avoided by assigning TD and DLD children more rigorously to the groups. My remark in the method is thus reflected in the results and in the discussion. - line 664: What other options can be proposed to complement or refine the identification of DLDs? You talk about it in the openings but maybe already put some indications in brackets.
Author Response
Response to Reviewer 3
Dear Reviewer
We would like to thank you very much for the careful reading and the plenty and very helpful comments to our manuscript. We have revised the paper substantially, and we are convinced that it improved a lot on the basis of your suggestions. Before we address the individual points of the review, we would like to start by summarizing the main changes:
- The reference standard is changed; classification is now based on referral and a language test
- The statistical analysis (mixed model) is changed according to the suggestions of the reviewers.
- The group comparisons are now based on non-parametrical tests because the data is not normally distributed.
- The manuscript has been proofread by a native speaker of English
Below we take the suggestion point by point.
The article focuses on the QU-NWRT (German). Studies on NWRTs are frequent and have already shown the potential of this kind of task to identify a DLD in a monolingual and bilingual population. However, The study is innovative in that it focuses on older children than in most NRWT research.
The structure of the article is adequate, easy for the reader. The introduced references are recent and provide a good overview of the subject and the studies on NWRT. The results are well presented. However, several general and specific comments have been incorporated to improve the article.
The weak points of the article are, in my opinion, the constitution and distribution of the TD and DLD groups, which then has an impact on the results of the study. Furthermore, the lack of inter-judge reliability for the analysis of the replicates is also a weak point. These elements can be further justified by the authors in the review.
Re Constitution and distribution of TD vs. DLD groups: We think that the new reference standard we used (referral + language test) captures this point, see also our response to you comment to line 348 below.
Re Interrater reliability: Thank you for the suggestion, which we will take up for future work. We did not calculate interrater reliability because we use broad orthographic instead of phonetic transcription and consider whole nonword accuracy. Coding this way reduces the likelihood to obtain different codings of accuracy. We do not think that the missing interrater reliability strongly weakens the findings because the results are very well in line with the expectations and earlier findings for younger age groups.
Title:
- of vs. auf ?
>> Thank you! The correct preposition is added.
Introduction :
- line 62: Which additional tools/measures can be used?
>> added:
“Both meta-analyses stress the need to use additional measures such as sentence repetition, lexical tasks, narration, and parental questionnaires, for more reliable identification of DLD and emphasize that the results…”
- line 125: Why are 5-9 year olds studied the most? Is this related to the age of diagnosis of DLD?
>> This has to do with a number of factors that - in our opinion - do not need to be mentioned: the clinical relevance of 5-and 6-year olds, but also with practical considerations and project-internal purposes. The authors themselves usually do not motivate the choice of the age groups in their articles.
- line 155: Is it always counted according to the number of correct phonemes? There seem to be differences according to the tasks and preferences of the authors (e.g., Tuller et al., 2018)
>> The differences metioned by Tuller et al (2018: 5) refer to differences in the scoring of two tasks: Sentence repetition and nonword repetition. Of course, phonemes are not relevant for SRep tasks, but our paper is not on sentence repetition. To take up your comment, we added an information:
“Whole item repetition corresponds to the target nonword in the number and order of phonemes, but the precise criteria for accuracy can differ across studies.”
- line 178: why not talk about the Youden indices (1950)? What are the differences with what you present in your paper?
>> Thank you for the suggestion. There are several possibilities to calculate the diagnostic values of a test. ROC analyses and Likelihood rations are currently the most common ways and, as we suppose, can be understood by most of the researchers and allows us to embed the results in the actual discussion. Both would be more difficult if we use the Youden indices.
- line 232: why is it interesting to look at data from 8-10 year olds? are there reasons? If so, they could be stated by the authors.
>> We are now more explicit regarding the rationale to consider this age group, see section ‚Rationale of the study and research questions’
- line 240: Are children referred to speech and language therapy between the ages of 8-10 necessarily referred for oral language? Isn't the priority given to learning and written language at this age? you explain something about it in the following lines, but the question is already being asked here.
>> Thank you for your question. I’ll add an information on this, but at a later point
“The classification as TD vs. DLD (= reference standard) is based on a) the referral to a speech-language intervention due to an oral language deficit, …”
In Germany, children are referred to SLTs due to oral language difficulties. This has to do with the separation between the health system (responsible for the diagnosis and therapy of spoken language difficulties) and education system (responsible for the diagnosis and training – but not therapy!!) of dyslexia and dysgraphia. Institutionally, written language difficulties are not considered as impairments (stange, but true). The system is very complicated and describing it goes beyond the scope of the paper. Despite of the regularities, it is possible that some DLD children have a written language deficit because spoken and written language difficulties often co-occur (see Footnote 5).
Method:
- line 328: why use SLI and DLD in a similar way? Focus on DLD and the CATALISE consensus of Bishop et al (2017).
>> Thank you I have added the reference to CATALIZE project and replaced SLI by DLD.
- line 333: Is the questionnaire used based on the PabiQ (Tuller, 2015)? If so, it should be cited.
Or what does your questionnaire add?
>> The questionnaire was earlier than the PABIQ because the project was already started before the PABIQ was available. I have added more information about the questionnaire. This is important because another reviewer asked for assessment in the L1. The early milestones (emergence of first words and word combinations) provide an indirect assessment (Tuller 2015, Boerma & Blom 2017); direct assessment was not possible or intended given the variety of L1’s in our sample.
“If the parents agree to participate, we contact them by telephone and conduct an interview based on a questionnaire. The questionnaire is developed for project-internal purposes and collects information about the child’s exposure to the L1 and L2, the language use at home and outside the family, socio-economic background (operationalized as mothers’ years of schooling). The questionnaire also inquiries about risk factors for early language development: attested hearing impairments, referral to a speech-language intervention, as well as spoken and written language impairments in the family (1st-grade relatives). Parents are also asked for attested cognitive, motor and social conspicuities. In the eL2 group, the age of onset to the L2 is defined as the age of the first systematic contact with German. No restrictions are made with respect to the L1, i.e. the eL2 learners acquire different L1s (see below for more details).”
- line 348: why only a test of syntactic comprehension was used? So your DLD children must have had receptive difficulties? What about the different domains?
A diagnosis is not made by assessing only one domain. Furthermore, a diagnosis of DLD is made when there are persistent difficulties and functional impacts, what about them? Indeed, it does not seem certain that the children classified as DLD in the research actually have DLD. For these reasons, it seems complex to then perform sensitivity and specificity calculations on children whose categorisation is not clear at the beginning.
>> Thank you for this important point. Unfortunately, the protocol of the project did not envisage further language tests (but a number of psycholinguistic experiments). In order to take up your point (which was made by reviewer 2 in a similar way), we confirm the initial diagnosis by a) referral to therapy, and b) early milestones.
- We now use the following reference standard: “Children are classified as TD if they were never referred to a speech-language therapy according to parental information and if they perform age-appropriately in the respective language test. Likewise, children are classified as DLD if they were referred to speech-language therapy and if they perform T < 40 in the language test. In the monolingual group, there is a perfect match of referral and result in the TROG-D, i.e. no over- or underdiagnosis. In the eL2 group, out of the original sample of n = 36 eL2 learners, referral and test results match in 33/36 (91.7%) children. The three remaining children all are underdiagnosed (i.e. not referred despite a poor test result). These three children are excluded from the analysis, resulting in n = 33 eL2 learners. “
Note that the now reference standard resulted in a re-classification of the eL2 learners and exclusion of three children. Monolinguals were not affected because there was a perfect match between referral and test.
- We now consider the early milestones as additional indicators of (non-)DLD. We tried to point out that these factors are associated with the abilities in the L1 in eL2 learners. Given that the age of onset is factually around age 3, the first words and word combinations should be produced in the L1, at least by the eL2-TD children but also by most of the eL2DLD children.
- line 447: why not conduct an interrater reliability?
>> Thank you for the suggestion, which we will take up for future work. Up to now, we did not calculate interrater reliability because we use broad orthographic instead of phonetic transcription and consider only whole nonword accuracy. Coding this way reduces the likelihood to obtain different codings of accuracy.
Material: if possible, it would be good to add the list of items (LI and LD) in the supplementary material or appendix
>> A list of items is given in the appendix.
Results:
- well presented, very clear
- but it might be a good idea to add the effect sizes in addition to the p. values
>> We did not compute effect sizes because this has to be done separately in the Mixed models, because it is not usual to provide them. The mean values presented in Table 4 provide indirect evidence that the effects are stronger in the language-dependent part. However, if you consider it as important, we can add effect sizes.
- line 501: I think it would be wise to explain how the cut-offs were calculated to make it easier for the reader to read and analyse.
>> Thank you for the suggestion. We understand your point, that unexperienced readers remain uncertain how the cutoffs are determined. In SPSS, the cut-offs are an output of the ROC analysis. Like for other statistical measures, we consider it as not important to explain how the cut-offs are determined by the program.
Discussion:
- line 558: and also in relation to the small number of studies that looked at LI items included in the meta-analyses.
>> We reformulated the paragraph slightly: “At first glance, the better diagnostic values found in the language-dependent part seem to contradict the results of the meta-studies that report better results for the language-independent items (Ortiz 2021) or no difference between the test parts (Schwob et al. 2021). This has to do with the construction of the tests and the small number of studies available so far. “
We hope that this formulation takes up your point.
line 657: The authors have well explained and justified the results obtained in terms of diagnostic accuracy.
However, in my opinion, this could have been partly avoided by assigning TD and DLD children more rigorously to the groups. My remark in the method is thus reflected in the results and in the discussion.
>> Thank you for this comment. The new reference standard takes it up.
- line 664: What other options can be proposed to complement or refine the identification of DLDs? You talk about it in the openings but maybe already put some indications in brackets.
>> Done: “Despite these promising results, the LITMUS-QU-NWR should always be accompanied by other measures (for example sentence repetition Abed Ibrahim and Fekete 2019; Tuller et al. 2018; Armon-Lotem and Meir 2016) in order to detect DLD with high confidence.”

Round 2
Reviewer 1 Report
General comments:
Lines 186 – 194 – I would like to see more clarification here leading to the conclusion of the paragraph that “the language-dependent part, reliably discriminates between TD and DLD in bilingual (and monolingual) children and that it does not penalize bilingual learners.” Is this conclusion based on the fact that internal codas and branching onsets are considered to be language-dependent?
Line 658: “Second, in line with previous studies, the TD-DLD differences are more pronounced in the language-dependent part than in the language-independent part” – what statistical result is this based on? I don’t see this spelled out in the results. The model with the LI items has a higher effect size than the LD items.
Why is it important to only analyze the LI items LD items separately? What happens to diagnostic accuracy if all items are included together under the same analysis?
Lines 728 - 737: strongly language-specific items? Are some language-specific items more language specific than others? Perhaps more nuance is required here. (also see next comment)
Pages 16-17: This part of the discussion appears to compare previous work with language-specific items using non-LITMUS tasks to the current findings and then previous work with language-specific items using a LITMUS-NWR tasks to the current findings, but it’s a bit hard to follow. Hence, the statement “The LITMUS-QU-NWR, however, follows different principles of construction.” (line 739) comes as a bit of a surprise. I would guide the reader through this a bit better.
Other editing:
Line 126: in the last years -> recently
128: are most studied -> are most frequently represented in studies involving….
178: that the impact of SES is neglectable -> negligible
Lines 481 – 503 (and elsewhere, such as when results from current or prior work are reported): I personally prefer the past tense here: The children are tested -> The children were tested
Tables 5 & 6: double checking formatting rules WRT numbers of decimals. Shouldn’t overall model values (R, AIC…) be reported?
FN 1: bottom effects -> floor effects
Author Response
Dear Reviewer 1,
thank you again for the careful reading and the helpful comments to our manuscript. We have revised the paper according to your suggestions and reply on your comments below. The editors asked for a version where changes are highlighted. We hope that you agree with our changes and the responses.
Below we take your suggestion point by point.
General comments:
Lines 186 – 194 – I would like to see more clarification here leading to the conclusion of the paragraph that “the language-dependent part, reliably discriminates between TD and DLD in bilingual (and monolingual) children and that it does not penalize bilingual learners.” Is this conclusion based on the fact that internal codas and branching onsets are considered to be language-dependent?
>> The conclusion is based on the findings summarized in the section. Maybe this was misleading because we didn’t start the summary with a new paragraph. We did so and clarified this point.
Line 658: “Second, in line with previous studies, the TD-DLD differences are more pronounced in the language-dependent part than in the language-independent part” – what statistical result is this based on? I don’t see this spelled out in the results. The model with the LI items has a higher effect size than the LD items.
>> Thank you for pointing this out. We agree with you that clarification is needed in this point but we don’t understand why the model with LI items has higher effect sizes than the LD part. To our knowledge, estimates and SE is not identical to effect sizes.
We added as sentence in the results section:
“The TD-DLD-differences are descriptively more pronounced in the LD compared to the LI part (monolingual: DLI = 14.1; DLD = 22.2; eL2: DLI = 17.1; DLD = 25.2).”
And made it clear in the discussion:
“Second, in line with previous studies, the descriptive results indicate stronger TD-DLD differences in the language-dependent part than in the language-independent part (monolingual: DLI = 14.1; DLD = 22.2; eL2: DLI = 17.1; DLD = 25.2). This is in line with previous findings (Somberg 2020; Grimm and Hübner accepted; Grimm and Schulz 2021; Abed Ibrahim and Fekete 2019; Scherger 2020; C. dos Santos and Ferré 2016).”
Why is it important to only analyze the LI items LD items separately? What happens to diagnostic accuracy if all items are included together under the same analysis?
>> We added a sentence:
“The two test parts are analyzed separately to see which part shows better diagnostic values for this age group.”
To take up your comment: If we take the whole test (LI + LD), we get better sensitivities for both groups, and a better AUC for the monolinguals, presumably because there are more items.
|
Mo (n = 36) |
eL2 (n = 36) |
|
LI + LD |
LI + LD |
AUC |
.973*** |
.938*** |
Cut-off (%) |
78.5 |
73.7 |
Sensitivity (% correct DLD) |
96.3 |
91.7 |
Specificity (% correct TD) |
88.9 |
77.8 |
***p < .001
We think that it is important to know which test part works better for our age group because the more appropriate part can be used in future research and – once norms are available – in clinical practice. Given limited time resources and the overall similarity of the items, it is better to test only 20 instead of 40 items. Note that 20 items are already many in comparison to other clinical NWR tests. This is the reason why we analyzed the parts separately.
Lines 728 - 737: strongly language-specific items? Are some language-specific items more language specific than others? Perhaps more nuance is required here. (also see next comment)
The sentence has been re-written:
“The TD-DLD differences in bilinguals can be less pronounced because even typically developing bilinguals often struggle with nonwords that are very similar to existing words a particular language (Gutiérrez-Clellen and Simon-Cereijido 2010; Boerma et al. 2015).”
And yes, the criterion ‘language-specific’ can be operationalized in many different ways: at the level of phonology by using language-specific segments or stress rules of a given language, or at the lexical level by changing a single sound of an existing word.
Pages 16-17: This part of the discussion appears to compare previous work with language-specific items using non-LITMUS tasks to the current findings and then previous work with language-specific items using a LITMUS-NWR tasks to the current findings, but it’s a bit hard to follow. Hence, the statement “The LITMUS-QU-NWR, however, follows different principles of construction.” (line 739) comes as a bit of a surprise. I would guide the reader through this a bit better.
We reformulated the paragraph:
“The LITMUS-QU-NWR, however, follows different principles of construction: The language-independent part is composed of cross-linguistically widely attested sounds, and the language-dependent part adds language-specific phonological complexity. Compared to other NWRs, the language-dependent part of the LITMUS-QU-NWR is still less word-like and shows little similarity to words in German. Hence, given that both test parts involve complex phonological structure and that children with DLD struggle with linguistic complexity, we expect that both parts discriminate between TD and DLD.”
Other editing:
Line 126: in the last years -> recently
>> changed to: With regard to group effects, a number of studies indicate that the LITMUS-QU-NWR provides a reliable tool to identify DLD in bilingual and monolingual children.
128: are most studied -> are most frequently represented in studies involving….
>> done
178: that the impact of SES is neglectable -> negligible
>> done
Lines 481 – 503 (and elsewhere, such as when results from current or prior work are reported): I personally prefer the past tense here: The children are tested -> The children were tested
>> Thank you. We changed to past tense if the action is finished.
Tables 5 & 6: double checking formatting rules WRT numbers of decimals. Shouldn’t overall model values (R, AIC…) be reported?
>> Thank you. We reduced Tables 5 and 6 to three decimals. We reported the usual statistical values, but we can add the other model values if you consider it as important.
FN 1: bottom effects -> floor effects
>> done

Reviewer 3 Report
This is a revised manuscript and I am satisfied with the revisions and have only a few minor comments.
Method: regarding the classification of TD and DLD children, this is better justified in the document. One question remains: Why not indicate the criterion of persistence of difficulties ? (e.g., children who have been diagnosed and followed for at least 3 months in therapy)
Author Response
Dear Reviewer 3,
thank you for the positive comment on the revised manuscript. We are pleased to hear that you are satisfied with the changes. Below you find our response to your final question.
This is a revised manuscript and I am satisfied with the revisions and have only a few minor comments.
Method: regarding the classification of TD and DLD children, this is better justified in the document. One question remains: Why not indicate the criterion of persistence of difficulties ? (e.g., children who have been diagnosed and followed for at least 3 months in therapy)
Thank you for the comment. In the literature, the persistence of DLD is normally not taken into account in connection with studies on diagnostic accuracy but of course this does not rule out the option to consider it as a criterion.
We didn’t ask the parents how long the children followed therapy. Another option to consider persistency is to look on the children’s performance in language tests at a later point. This is not possible because this subgroup has been tested cross-sectionally (younger participants of the project were also tested longitudinally). We will keep your comment in mind for future work.
